# On the Power of Louvain in the Stochastic Block Model

Vincent Cohen-Addad[1], Adrian Kosowski[2], Frederik Mallmann-Trenn[3], and David Saulpic[4]

[1]Google Zürich, Switzerland
[2]NavAlgo, France
[3]Sorbonne Université, UPMC Univ Paris 06, CNRS, LIP6, France
[4]King's College London, UK

## Abstract

A classic problem in machine learning and data analysis is to partition the vertices of a network in such a way that vertices in the same set are densely connected and vertices in different sets are loosely connected.

In practice, the most popular approaches rely on local search algorithms; not only for the ease of implementation and the efficiency, but also because of the accuracy of these methods on many real world graphs. For example, the Louvain algorithm – a local search based algorithm – has quickly become the method of choice for clustering in social networks. However, explaining the success of these methods remains an open problem: in the worst-case, the runtime can be up to $\Omega(n^2)$, much worse than what is typically observed in practice, and no guarantee on the quality of its output can be established.

The goal of this paper is to shed light on the inner-workings of Louvain; only if we understand Louvain, can we rely on it and further improve it. To achieve this goal, we study the behavior of Louvain in the famous two-bloc Stochastic Block Model, which has a clear ground-truth and serves as the standard testbed for graph clustering algorithms. We provide valuable tools for the analysis of Louvain, but also for many other combinatorial algorithms. For example, we show that the probability for a node to have more edges towards its own community is $1/2 + \Omega(\min(\Delta(p-q)/\sqrt{np}, 1))$ in the SBM$(n, p, q)$, where $\Delta$ is the imbalance. Note that this bound is asymptotically tight and useful for the analysis of a wide range of algorithms (Louvain, Kernighan-Lin, Simulated Annealing etc).

## 1 Introduction

Local search algorithms are widely-used in machine learning and data analysis, to extract information or optimize models. Among the most classic examples are Gradient Descent for tuning neural networks, Lloyd's method and Expectation-Maximization (EM) for clustering, unsupervised learning and statistical inference. However, understanding the practical success of local search algorithms through a theoretical analysis remains a major open problem. Proving guarantees on the quality of the local optima found by the algorithm and the required running time remain notoriously hard problems. For most of the above mentioned methods it is possible to construct adversarial examples that lead to highly sub-optimal local optima or induce very slow convergence. Nonetheless, many of these worst-case examples are contrived and highly unlikely to arise in real-world scenarios. Therefore, if one seeks to understand the success of local search algorithms, one must go *beyond the worst-case* scenario. This path has been recently explored for various algorithms and numerous papers have recently shown the power of gradient descent, EM, Lloyd's method in various specific contexts.

An illustrative example of the discrepancy between the success of local search techniques versus its theoretical understanding is the case of graph partitioning. Consider the problem where one is given a graph $G$, and asked to partition it into subgraphs, each of which exhibiting a higher density of edges within the subgraph than towards the rest of the graph. For this problem, the power of local search algorithms was first materialized by simulated annealing heuristics. In the early 70s, Kernighan and Lin [26] presented a simple local search procedure for computing a balanced cut[1] of a graph of small size. The heuristic quickly became a standard tool for VLSI design and is still part of various packages [32]. More recently, the success of the Louvain algorithm [7] for extracting information from social networks, knowledge or similarity graphs has shown that despite a flurry of new techniques, local search algorithms remain the most popular heuristics. However, from a theoretical standpoint, designing approximation algorithms for graph partitioning objectives such as modularity, sparsest cut, bisection, or multicut is a major challenge: under some popular complexity assumptions such as the unique game conjecture or $P \neq NP$, there is no constant factor polynomial-time approximation algorithms for the above problems.

**Modularity and the Louvain algorithm.** Introduced in 2008 and designed to detect communities in social networks, the Louvain heuristic has received more than 11400 citations over the last 10 years [19] and is now the method of choice for clustering similarity graphs (see for example the extensive analysis of Lancichinetti and Fortunato [27]). The algorithm is simply a slight refinement of a local search algorithm which aims at optimizing the *modularity* of the current clustering (see Equation 1 and a more detailed presentation of the Louvain algorithm in Section 2). More interestingly, this algorithm is recognized to produce a *good* clustering very fast, outperforming most of the other clustering methods.

Even though this heuristic is now widely used, it is known that it may output arbitrarily bad partitions (in terms of modularity) for some adversarial examples. Even more surprising is the fact that the worst-case running time of the algorithm is $\Omega(n^2)$, a prohibitive running time in practice, but experiments show that it often terminates after $O(n \operatorname{polylog}(n))$ operations.

Quite surprisingly, our understanding of the success of this heuristic is very poor: no guarantees on the quality of the solution output by Louvain, even for some simple scenarios, have been established. We thus ask: what is the structure of Louvain's solution for real-world graphs?

A natural setting for providing a beyond-worst-case analysis of these local search algorithms is through the classic Stochastic Block Model (see formal definition in SuppMat A) which exhibits a *clear ground-truth clustering* and which has been used to provide a beyond-worst-case-analysis framework in a large number of works.

## 1.1 Our Results

We focus on the classic Stochastic Block Model with two communities, namely the graph consists of two communities, each consisting of $n$ nodes, and the probability of observing an edge between two nodes of the same (resp. different) community is $p$ (resp. $q$). We refer the reader to Section 2 for formal definition of the above concepts and the Louvain algorithm. Our results are two-fold. We show, for a large range of parameters, that Louvain recovers the hidden partition and that it converges rapidly.

We first show that if Louvain is initialized properly, namely with an equal-size two-partition with *imbalance* $\Delta$, i.e.: where some part contains $n/2 + \Delta$ vertices of a given community, then it converges in $O(n)$ steps to the correct clustering with high probability assuming $\frac{p-q}{\sqrt{p}} \geq c\sqrt{n \log n}/\Delta$, for some constant $c$. Interestingly, this bound is near-optimal, namely close to the information theoretic threshold $\frac{p-q}{\sqrt{p}} \geq \sqrt{\log n/n}$ up to constant factors, if $\Delta \geq n/c'$ for some constant $c'$.

**Theorem 1.1** (Warm Start). *Let $\Delta > 0$. Consider a graph $G \sim \mathrm{SBM}(n, p, q)$. Then, there exists a constant $c$ such that, with high probability,* LOUVAIN *initialized with a partition of imbalance $\Delta$ recovers the partition $\{V_1, V_2\}$ in $O(n)$ rounds, if $\frac{p-q}{\sqrt{p}} \geq 200 \frac{\sqrt{\log n}}{\sqrt{\Delta}} \max\left(1, \frac{\sqrt{(n/2-\Delta)}}{\sqrt{\Delta}}\right)$.*

We then show that even when Louvain is initialized with a random equal-size two-partition, it converges to the correct clustering with high probability after only $O(n)$ steps, provided that $p-q/\sqrt{p} \geq 100n^{-1/6+\varepsilon}$.

**Theorem 1.2** (Cold Start). *Consider a graph $G \sim \mathrm{SBM}(n, p, q)$ and assume $\frac{p-q}{\sqrt{p}} \geq 200n^{-1/6+\varepsilon}$, for some $\varepsilon > 0$. With hight probability,* LOUVAIN *algorithm recovers the partition $\{V_1, V_2\}$ within $O(n)$ rounds.*

To prove these theorems, we provide valuable tools for the analysis of Louvain, but also for a wide range of combinatorial algorithms. For example, we show that the probability for a node to have more edges towards its own community is $1/2 + \Omega(\min(\Delta(p-q)/\sqrt{np}, 1))$ in the $\mathrm{SBM}(n, p, q)$, where $\Delta$ is the imbalance. Note that this bound is asymptotically tight and useful for the analysis of other local-search based algorithm such as the aforementioned Kernighan-Lin, Simulated Annealing etc.

As a side product of our techniques we also obtain bounds for MAJORITY, which is a simpler version of Louvain, where a node simply moves to the part to which it has the most number of edges. We can show that for $p - q \geq 1/n^{1/4}$, MAJORITY recovers the optimal partition in $O(n^2 p)$ steps, which is linear in the graph size. In comparison, the state-of-the-art, [9] showed that MAJORITY if $p - q \geq 1/n^{1/4}$ in dense graphs, namely when the number of edges is $\Omega(n^2)$. In contrast to their techniques, ours does not have any requirement on the density of the graph. Another drawback of their analysis is that it does not imply that the convergence time would be subquadratic. Here we show that it is in fact linear.

## 1.2 Comparison to Previous Work

Understanding the power of local search for graph cut problems has always been of high interest for the research community. The classic *majority* algorithm has been studied since the work of Kernighan and Lin [26]: the algorithm maintains a two-partition of the graph and swap a node from one side to the other if it has more neighbor in the latter. The research community has first taken an important step towards understanding local search algorithms in the Stochastic Block Model through the work of Jerrum and Sorkin [24, 25] on the metropolis algorithm for graph bisection. They showed that in the Stochastic Block Model with 2 communities, the metropolis algorithm (simulated annealing at some specific fixed temperature) recovers the optimal bisection if $p - q \geq 1/n^{1/6}$ after $O(n^2)$ steps. This was later improved by Carson and Impagliazzo [9] who showed that the standard local search algorithm also recovers the optimal partition if $p - q \geq 1/n^{1/4}$ in dense graphs, namely when the number of edges is $\Omega(n^2)$. However, this result is unsatisfactory in two aspects: first, the proof critically relies on the number of edges being $\Omega(n^2)$, which is for this type of algorithms arguably a strong assumption since the information per node is much higher than in a sparse regime. More importantly, the result did not address the running time of the algorithm (i.e. the convergence time of the process). Thus, in addition to the first analysis of LOUVAIN, our results also improve upon the work of Carson and Impagliazzo on the analysis of the Majority algorithm by addressing sparser regimes, obtaining a strong bound on the running time. More recently, Boumal [8] showed that simulated annealing at the "correct" temperature recovers the correct partition nearly-optimality (namely up a constant factor of the information theoretic threshold). However, the temperature should be set as a function of the model parameters and so this algorithm remains far from practical. More recently, Chin, Rao and Vu [10] and Yun and Proutière [34] have designed local-search-based algorithms that aim at improving a solution obtained via spectral method. Both proofs assume that the initial partition given to the local search algorithm only missclassifies a very tiny fraction of the vertices (only $O(1/p)$ vertices are misclassified in [34], $O(n/10)$ in [10] – Note also that [10] considers an algorithm that is designed to avoid most of the technical issues encountered when analysing local search methods since it at each step it works with 'fresh' edges for which the randomness has not been revealed). Those results are therefore far from addressing the cold start setting, which is the most challenging and interesting for the analysis of local-search heuristics, while our results on the warm start setting are strictly more general.

From a technical standpoint, an important challenge that our work addresses is handling the randomness of the graph through successive local search steps. This is a key step when analysing a local search algorithm since it is particularly hard to deal with the dependencies created by the algorithm, which considers every edge many times. To the best of our knowledge, previous work tackled this

issue by carefully designing their algorithms. This is not possible to do when analyzing LOUVAIN, and we therefore must develop new tools. On the one hand, the existing local-search algorithms of [25, 11, 9] are designed such as to avoid this dependency issue, by using at every step "fresh" edges, for which the randomness has not been revealed until that step. On the other hand, the techniques developed in the series of work dedicated to the Stochastic Block Model mentioned above relies mostly on SDP or spectral graph theory, and do not seem to apply to local-search heuristic. From a performance standpoint, those algorithms recover the partition when $p-q/\sqrt{p} = O(\sqrt{\log n/n})$.

There is a large body of other work on the Stochastic Block Model and describing it is beyond the scope of this paper. The interested reader may look into the survey of Abbe [1]. The precise understanding of what can be recovered as a function of $p$ and $q$ in the Stochastic Block Model is due to Abbe et al. [2] and Mossel et al. [30]. They prove that recovery is possible if and only if $p-q/\sqrt{p} > 2\sqrt{\log n/n}$. Classic results encompass the fundamental result of McSherry [29], the augmentation algorithm of Condon and Karp [11]. Iterative methods [16, 28, 35, 17], semi-definite programming [20, 21, 5, 13, 14] and spectral algorithm [1] have been investigated under the Stochastic Block Model. Perhaps, more closely related results are the recent advances on the analysis of the Belief Propagation (BP) algorithm, a much more evolved message-passing than the standard MAJORITY. Some algorithms, based on BP or variants of BP (such as the linearized acyclic BP) have been shown to recover the ground-truth output in the Stochastic Block Model as well [4, 3, 12]. Nonetheless, we believe that these works, while of high importance for the study of BP algorithms do not allow to shed light on simpler heuristics, such as MAJORITY or LOUVAIN which are widely-used in practice and also reasonable models of local-decision dynamics.

### 1.3 Roadmap

In Section 2 we introduce the algorithms. A formal definition of the Stochastic Block Model can be found in SuppMat A together with some additional notations. In Section 3, we study the behaviour of LOUVAIN when initialized with a large imbalance, and prove Theorem 1.1. In Section 4, we study the algorithm initiated with a random cut, and show Theorem 1.2. All proofs can be found in the supplementary material.

## 2 Preliminaries and Notations

The formal definition of the Stochastic Block Model can be found in SuppMat A. In short, there are two communities each with $n$ nodes. Two nodes from the same community are connected with probability $p$ and nodes from different communities are connected with probability $q$. The goal is to recover the two communities.

**The LOUVAIN Algorithm**  We now describe the local-search algorithm LOUVAIN ([7]). Although this article focuses on the case with two communities, LOUVAIN is more general and we define it for more communities here. It is a local search technique that aims at finding a partition of the vertices of a given graph that maximizes the *modularity*. For any partition $P = (P_1, \ldots, P_\ell)$, the modularity of $P$ is defined as

$$M(P) = \frac{1}{2m} \sum_{i=1}^{\ell} \sum_{u,v \in P_i} \left( 1_{(u,v)} - \frac{\deg(u) \cdot \deg(v)}{2m} \right), \tag{1}$$

where $1_{(u,v)}$ is 1 if and only if there is an edge between vertices $u$ and $v$, $\deg(u) = \sum_v 1_{(u,v)}$, $2m = \sum_u \deg(u)$. For a vertex $u$, we let $P(u)$ be its part in the partition $P$. The Louvain local dynamic is defined as follows. Consider a partition $P = (P_1, \ldots, P_\ell)$. For each vertex $u$, define $P^{u,i}$ to be the partition where $u$ is removed from $P(u)$ and added to $P_i$. Define $Q_{u,i}$ as the modularity of $P^{u,i}$ minus the modularity of $P$ and let $i_u^*$ be $\arg\max_i Q_{u,i}$, breaking ties arbitrarily. We say that $Q_{u,i}$ is the swap value for $u$.

The Louvain algorithm consists of successive steps, where each step is performed as follows. Given a partition $P$, the algorithm considers all the vertices $u$ such that $P(u) \neq P_{i_u^*}$, picking a pair $u, v$ such that $P(u) = P_{i_v^*}$ and $P(v) = P_{i_u^*}$ at random and then defining a new partition $P'$ obtained by removing $u$ from $P(u)$ and adding it to $P_{i_u^*}$ and removing $v$ from $P(v)$ and adding it to $P_{i_v^*}$ Then,

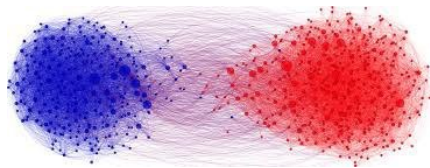
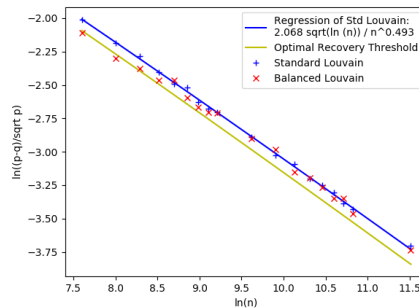

(a) The figure depicts a graph generated by the Stochastic Block Model. The nodes within the communities are densely connected and nodes of different communities are less densely connect.

(b) The points represent, for a given number of points $n$, the smallest value of $\frac{p-q}{\sqrt{p}}$ for which Louvain (resp. Standard Louvain) succeeds. More details are given in Section 5.

the algorithm performs the next step on partition $P'$. The algorithm stops when the partition $P$ is such that $P(u) = P_{i_u^*}$, for all $u$.

For $k = 2$ communities, this is very similar to the classic Hillclimbing procedure considered in [9] and the Metropolis algorithm as temperature 0 considered in [24].

The algorithm we analyze, Balanced Louvain, is a slight modification of the algorithm above (which we call Standard Louvain) in three ways.

1. First, Balanced Louvain starts with a random equi-sized partition $P = (P1, P2)$, whereas Standard Louvain starts with $2n$ parts each containing one node.

2. Second, Standard Louvain moves only one node at a time, whereas Balanced Louvain swaps nodes, to maintain balanced clusters. More precisely, Balanced Louvain select one node with positive swap value in each part, and moves it to the other part.

3. Third, once a local optimum is reached, Standard Louvain merges the nodes of a cluster together and proceeds. Balanced Louvain simply stops when a local optimum is reached.

We now make a case for these adaptations are justified. 1) At the beginning Louvain will quickly reduce clusters until there are only two left. Essentially, Louvain does not encounter any local optima when the number of clusters is strictly larger than two. 2) We adopted this variant of Louvain to avoid having to keep trace of the size imbalance between communities during the process. It can be shown with random walk argument that, assuming 1), the size imbalance stays negligible: This is done in SuppMat F. 3) By considering our simpler procedure, without any contraction, we actually show that for the SBM, the hierarchy has a single level, and the algorithm does not need to escape local optimum. Just like one would hope.

To further justify our adaptations, we show experimentally that our variant, Balanced Louvain, performs just as well as Louvain (see Figure 1b and Section 5).

## 3   Warm Start

In this section we consider a graph $G \sim \mathrm{SBM}(n, p, q)$ and a partition $V = (P_1, P_2)$ with imbalance $\Delta$. For any $i \in \{1, 2\}$, we refer to the part that contains the larger number of vertices of community $i$ as the *home* of $i$ and we refer to this part as HOME $i$. Namely, HOME $1 = \mathrm{argmax}_{U \in \{P_1, P_2\}} |U \cap V_1|$ and HOME $2 = \mathrm{argmax}_{U \in \{P_1, P_2\}} |U \cap V_2|$. We say that a vertex is *good* if it is of community $i$ and not in HOME $i$. A vertex is *bad* if it is of community $i$ and in HOME $i$. This section is dedicated to the proof of Theorem 1.1. The proof idea is built around showing the following property $\mathcal{P}$. For any given cut with large imbalance $\Delta$, there is no large subset of nodes whose sum of swap values is negative. If the sum of swap values is non-negative, this also means that among the good nodes, there can only be very few with a negative swap value. Therefore, $\mathcal{P}$ implies that most of the good nodes would move to their HOME if chosen. Similarly, we can show that most of the bad nodes prefer to stay in their HOME. Putting both of these facts together, we get that the imbalance is likely to increase after one round. We can show that the imbalance 'performs' a biased random walk and will therefore quickly increase to a size of $n/2$ implying convergence.

One of the main technical challenges of proving $\mathcal{P}$ is the dependencies among the nodes: the swap values of the nodes are correlated due to mutual edges. However, this dependency is weak and we can use Theorem A.1 to obtain strong concentration bounds. So strong, that we can take a Union bound over all cuts with large imbalance taking care of revealed randomness in previous steps. The bulk of the proof is captured by the following lemma.

**Lemma 3.1** (proved in SuppMat B). *Consider a graph $G \sim \mathrm{SBM}(n, p, q)$ on $n$ vertices and let $0 \leq \Delta < n/2$. Assume $\frac{p-q}{\sqrt{p}} \geq 200 \frac{\sqrt{\log n}}{\sqrt{\Delta}} \max\left(1, \frac{\sqrt{(n/2-\Delta)}}{\sqrt{\Delta}}\right).$*

*Fix a partition $(S, V \setminus S)$ with imbalance $\Delta < n/2$, the following holds with probability at least $1 - 3\exp(-5(n/2 - \Delta)\log n)$: The number of good vertices with positive swap values is at least $(2/3)(n/2 - \Delta)$ and the number of bad vertices with positive swap values is at most $(n/2 - \Delta)/3$.*

Assuming this lemma, we can prove the main proposition in SuppMat B. Briefly, Lemma 3.1 is used to show that with probability $1 - O(1/n^2)$ all cuts are *improving*, i.e. the probability of increasing the imbalance is at least $2/3$. The imbalance is therefore a random walk on $\mathbb{N}$, with probability $2/3$ of increasing: the time to reach $n$ is thus $O(n)$ with probability $1 - O(1/n^2)$, which concludes the lemma.

The rest of this section is dedicated to the proof of Lemma 3.1. Our strategy is to consider the sum of swap values of a big enough set of vertices $S_0$. Since this is in expectation way larger than the swap value of a single vertex – it is $|S_0|\Delta(p-q)$ for a set of good vertices – Chernoff bounds allows to show concentration with way higher probability. The first part of Lemma 3.2 shows that the sum of swaps values of at least $\frac{1}{3}(n/2 - \Delta)$ bad vertices must be negative. This is used as follows: let $S_0$ be the set of bad vertices with positive swap value. If $S_0$ had size bigger than $\frac{1}{3}(n/2 - \Delta)$, it would contradict Lemma 3.2. Hence, the number of bad vertices with positive swap value is at most $\frac{1}{3}(n/2 - \Delta)$.

Similarly, the second statement of the lemma shows that a big enough group of good vertices must have positive total swap value. This implies as well that, if $S_0$ is the set of good vertices with negative swap value, $S_0$ must have size smaller than $\frac{1}{3}(n/2 - \Delta)$. Hence, since there are $n/2 - \Delta$ good vertices, there must be at least $\frac{2}{3}(n/2 - \Delta)$ good vertices with positive swap value.

**Lemma 3.2** (proved in SuppMat B). *Consider a graph $G \sim \mathrm{SBM}(n, p, q)$ on $n$ vertices and assume that $\frac{p-q}{\sqrt{p}} \geq 200 \frac{\sqrt{\log n}}{\sqrt{\Delta}} \max\left(1, \frac{\sqrt{(n/2-\Delta)}}{\sqrt{\Delta}}\right).$*

*Fix a partition $(S, V \setminus S)$ with imbalance $\Delta < n/2$, and let $S_0 \subseteq S$ be a set of vertices of size $\frac{1}{3}(n/2 - \Delta)$.*

- *If $S_0$ consists only of bad vertices, then the sum of swap values of the vertices in $S_0$ is at most $-|S_0|\Delta(p-q)/2$ with probability at least $1 - 3\exp(-5(n/2 - \Delta)\log n)$.*

- *If $S_0$ consists only of good vertices, then the sum of swap values of the vertices in $S_0$ is at least $|S_0|\Delta(p-q)/2$ with probability at least $1 - 3\exp(-5(n/2 - \Delta)\log n)$.*

The proof of Lemma 3.1, presented in SuppMat B simply uses the previous lemma as explained previously.

## 4  Cold Start

We start by giving the intuition. The proof has two parts. In the first part (Section 4.1), we assume that we start with a graph with *fresh* randomness (i.e.: nothing about the random process generating the edges has been revealed so far), and a random partition into 2 parts having imbalance at least $\Delta$. That is, we assume that we have $n/2 + \Delta$ nodes of community 1 and $n/2 - \Delta$ nodes of community 2 in the first part of the partition, and that we draw edges in the graph according to the Stochastic Block Model. Then, we show that the probability of a node $u$ having more edges to its HOME is at least $p' = 1/2 + 0.018 \cdot \min\left\{\Delta(p-q)/\sqrt{np(1-p)}, 1\right\}$. Note that this term is up to constants in the second-order term tight and improves on the result of [11] that did not have the factor $1/\sqrt{p(1-p)}$ in the second-order term. For our results, which allows $p$ to be very small, this term is vital. To obtain it, we use Esseen's inequality together with coupling arguments. From this, we deduce that a large fraction of the node have more edges to their HOME than to the other part; which in turn implies that

Louvain has a good probability of moving one node to its HOME and improve the imbalance. When the imbalance is $\Omega(n/\log^2(n))$ we can appeal to the warm start result. The challenge is thus to show that the following property $\mathcal{P}$ holds: The fraction of nodes that has more edges to its HOME than to the other community is at least $1/2 + \Omega\left(\min\left\{\Delta(p-q)/\sqrt{np(1-p)}, 1\right\}\right)$.

In the second part (Section 4.2), we aim at proving that the probability that property $\mathcal{P}$ holds for all the cuts encountered by the algorithm is indeed high. To do so, we proceed as follows. From Section 4.1 we know that property holds for a random cut of a fresh graph with at least constant probability but this is not good enough because after one iteration of Louvain (i.e.: swapping one vertex from one side to the other) some of the randomness of the graph has been revealed. Hence, the cut reached after one iteration cannot be considered to be on a fresh graph, preventing us from directly applying the above result. We then argue that $\mathcal{P}$ has exponentially small probability of not happening and [2] applying a counting argument, we show in lemma Lemma 4.7 that the property $\mathcal{P}$ holds w.h.p. for all of the partitions encountered by Louvain during the first $n/\log^2(n)$ steps. The counting argument is arguable simple and essentially states that the number of partitions that can possibly be encountered after $t$ steps is at most $2^{t\log n}$ but the probability of a random cut to be problematic is exponentially smaller than that number. Hence, the probability that the initial random cut could lead to a cut for which property $\mathcal{P}$ does not hold is exponentially small. From there on we can use Markov chain theory to argue that the behavior of the imbalance $\Delta$ can be modeled by a biased random walk and will quickly increase to a size that is covered by the warm start regime (Section 3). From thereon, the process quickly converges and we obtain the hidden partition.

## 4.1 The probability of Improving the Cut

The goal of this section is to prove Lemma 4.1, which in substance states that for a graph with imbalance $\Delta$ and fresh randomness, the probability of a node $u$ having more edges to its HOME is at least $p' = 1/2 + 0.018 \cdot \min\left\{\Delta(p-q)/\sqrt{np(1-p)}, 1\right\}$ (simplified).

To this end we introduce some notation. Let $X_i(u)$ denote the number of edges from $u$ to part $i$, for $i \in \{1,2\}$. Each $X_i(u)$ decomposes as the sum of two binomials with different parameters. These variables encapsulate the movement of $u$: $u$ goes to community $i$ such that $X_i(u) = \max\{X_1(u) + L_1, X_2(u) + L_2\}$, where $L_1$ and $L_2$ are the Louvain terms. We seek to calculate the probability that a given $X_1(u)$ is larger than $X_2(u)$, by at least $|L_1| + |L_2|$ (note that $L_1$ can be negative). To do this we use Esseen's inequality to show that $X_1(u)$ and $X_2(u)$ are distributed very similarly to $Y_1(u)$ and $Y_2(u)$, where $Y_i(u), i \in \{1,2\}$ is the Gaussian equivalent of $X_i(u)$ (see Lemma 4.2). We then introduce ideal Gaussians $\{Z_i(u)\}_{i\in\{1,2\}}$ coupled with $\{Y_i(u) + L_i\}_{i\in\{1,2\}}$. The ideal Gaussians $\{Z_i(u)\}_{i\in\{1,2\}}$ allow us to use some symmetry properties enabling us to bound the probability that node $u$ goes to the other part) tightly up to a constant in the second-order term (Lemma 4.1).

We now give the formal definitions. For a given vertex $u$ with COMMUNITY$(u) = 1$, we define the following random variables corresponding to the number of edges $u$ has to part 1
$$X_1(u) \sim B(n/2 + \Delta, p) + B(n/2 - \Delta, q) \quad \text{and} \quad X_2(u) \sim B(n/2 - \Delta, p) + B(n/2 + \Delta, q)$$

As mentioned before, the goal of this section is to prove the following lemma, which gives a lower bound on the probability of improving the cut. We will use $\mathrm{L}^*$ which will be a bound on the Louvain terms, that is $|L_1| + |L_2| \leq \mathrm{L}^*$.

**Lemma 4.1.** *Assume* $|L_1| + |L_2| \leq \mathrm{L}^*$*. Then,* $\mathbb{P}\left[X_1 \geq X_2 + \mathrm{L}^*\right]$ *is at least*

$$1/2 + 0.018 \cdot \min\left\{\frac{\Delta(p-q)}{\sqrt{np(1-p)}}, 1\right\} - \frac{1}{2\Delta(p-q)} - \frac{\mathrm{L}^*}{2\sqrt{(n/2 - \Delta)p(1-p)}} - 4\sqrt{\frac{2}{np(1-p)}}$$

In order to prove the lemma, we define the normally distributed random variables corresponding to $X_1(u)$ and $X_2(u)$.

$$Y_1(u) \sim \mathcal{N}((\frac{n}{2} + \Delta)p + (\frac{n}{2} - \Delta)q, \sigma_1) \text{ and } Y_2(u) \sim \mathcal{N}((\frac{n}{2} - \Delta)p + (\frac{n}{2} + \Delta)q, \sigma_2) \quad (2)$$

where $\sigma_1 = (n/2+\Delta)p(1-p)+(n/2-\Delta)q(1-q)$ and $\sigma_2 = (n/2-\Delta)p(1-p)+(n/2+\Delta)q(1-q)$. We will define two normally distributed random variables $Z_1(u)$ and $Z_2(u)$ which we will use to argue that the Louvain term does not influence the outcome. We will assume they have the same law as $Y_j$. Define $Z_j \stackrel{d}{=} Y_j$ for $j \in \{1, 2\}$ .[3]

In the following, we will focus on a particular vertex $u$, assuming w.l.o.g that COMMUNITY$(u) = 1$. Hence, we drop the parenthesis from the variables $X, Y$ and $Z$. We show that the binomials $X$ behave very similarly to their Gaussian counterparts $Y$, which, together with the Louvain term, we will couple with $Z$. We then use these similarity between $X$ and $Z$ to prove Lemma 4.1.

In a first step, we relate the Binomials $X_j$ and the Gaussians $Y_j$.

**Lemma 4.2** (proved in SuppMat C). *Assume* $|L_1| + |L_2| \leq L^*$. *We have for all* $i, j$
$$\left| \mathbb{P}\left[ X_1 \geq X_2 + L^* \right] - \mathbb{P}\left[ Y_1 \geq Y_2 + L^* \right] \right| \leq 4\sqrt{\frac{2}{np(1-p)}}.$$

Note that $X_1$ and $X_2$ are Binomial random variables, i.e., the sums of Bernoulli random variables. In the proof, we use Esseen's inequality (Theorem E.1) to convert the Binomials to Gaussians.

In a second step, we relate between the Gaussians $Y_j$ and $Z_j$.

**Lemma 4.3** (proved in SuppMat C). *Assume* $|L_1| + |L_2| \leq L^*$. *We have*
$$\left| \mathbb{P}\left[ Y_1 > Y_2 + L^* \right] - \mathbb{P}\left[ Z_1 > Z_2 \right] \right| \leq \frac{L^*}{2\sqrt{\mathrm{Var}[Y_2]}}.$$

In the proof, we show that there exists a coupling such that $\forall j, L \leq L^* : \mathbb{P}\left[ Y_j + L = Z_j \right] \geq 1 - \frac{L}{2\sigma_2}$. We do this by bounding the total variation distance between the distributions $Y_j$ and $Z_j$.

Carefully analyzing Gaussians allows us prove the following lemma which gives bounds that are tight up the constant in the second-order term.

**Lemma 4.4** (proved in SuppMat C). *We have* $\mathbb{P}\left[ Z_1 > Z_2 \right] \geq 1/2 + 0.018 \cdot \min\left\{ \frac{\Delta(p-q)}{\sqrt{np(1-p)}}, 1 \right\} - \frac{1}{2\Delta(p-q)}$.

We now have all parts required to prove Lemma 4.1, which we do in the supplementary material. Note that choosing $L^* = 1$ provides the same guarantees we get for Louvain for MAJORITY. The proof can be found in SuppMat C.

## 4.2 From Imbalance $\sqrt{n}$ to $\Omega(n/\log^2 n)$

We now use Lemma 4.1 to show that the imbalance rapidly grows. For this, we start with a random cut, which can be assumed to have imbalance $\sqrt{n}$. This, together with Lemma 4.5 that bounds $L^*$, allows us to compute the probability that a random positive swap improves the cut. We conclude the section in Lemma 4.7, comparing the imbalance with a random walk to show its growth.

The first step stems from the fact that, with constant probability, the imbalance of a random cut is more than $\sqrt{n}$. The probability can be boosted by repetition, since no randomness of the edges is revealed. The next step is to bound the term $L^*$.[4] This is captured in the following lemma.

**Lemma 4.5.** *[proved in SuppMat C] Given a random cut with imbalance $\Delta$, there exists a constant $c$ such that, with probability $1 - 2\exp(-\frac{\Delta^2(p-q)^2}{cp})$, it holds that for all vertex $u$, $|L(u)| \leq \Delta(p - q)/100$.*

We now combine Lemma 4.5 with Lemma 4.1 to get the probability that a swap vertex chosen by the algorithm is good – and note *good* this event. We note *positive* the event that the vertex chosen

by the algorithm has positive swap value. Recall that a *good* vertex is a vertex that is not in its home, so that we would like to swap to the other side.

**Lemma 4.6.** *[proved in SuppMat C] Assume $\frac{p-q}{\sqrt{p}} \geq 200n^{-1/6+\varepsilon}$. Fix some imbalance $\Delta \in [\sqrt{n}, n/\log^2 n]$. Depending on the size of $\Delta$ the following holds. There exists constants $c_1, c_2$ such that:*

1. *For $\Delta(p-q)/\sqrt{n} \leq 1$, we have w.p. $1 - \exp\left(-\frac{\Delta^2(p-q)^2}{100}\right)$ that $\mathbb{P}[\,good \mid positive\,] = 1/2 + c_1\frac{\Delta(p-q)}{n}$, for some constant $c_1$.*

2. *For $\Delta(p-q)/\sqrt{n} > 1$, we have w.p. $1 - \exp\left(-\frac{c_2^2 n}{4}\right)$ that $\mathbb{P}[\,good \mid positive\,] = 1/2 + c_2$, for some constant $c_2$.*

Now that we know that the algorithm has a good probability to increase the imbalance, we can formalize the convergence with a random walk argument:

**Lemma 4.7** (proved in SuppMat C). *Assume $\frac{p-q}{\sqrt{p}} \geq 100n^{-1/6+\varepsilon}$. Then, after $O(n/\log n)$ steps of the algorithm, we have that $\Delta = \Omega(n/\log^2 n)$ with probability $1 - 1/n$.*

### 4.3 From Imbalance $\Omega(n/\log^2 n)$ to convergence

We can finally conclude the proof of Theorem 1.2, combining Lemma 4.7 and the results of Section 3, Theorem 1.1. The proof can be found in the supplementary material.

## 5 Experiments

We experimentally evaluated the performances of Louvain in the SBM. For the experiment we used the standard Louvain and the vertex-swapping version that we analyze. Our implementations builds on the Louvain implementation of Guillaume [23]. In order to generate the graphs efficiently, we devised a method that draws the the edges from the correct probability distribution in $O(m)$-time instead of $O(n^2)$ time, where $m$ is the number of expected edges $\approx n^2(p+q)$.

In our experiments, we set $q = p/2$. The plotted curve is the smallest value of $p-q/\sqrt{p} = \sqrt{p}/2$ for which the algorithm recovers the ground truth at least 8 times out of 15 trials on different random graphs. We use a log-log scale for plots. We added to them the curve $2.068 \log n/n^{0.493}$, found with non-linear least squares to fit Louvain's performances curve.

We make the two following observations. First, the exponent in our analysis $(1/6)$ does not seem tight. It is worth noting that $0.5$ is optimal, as it is known that $p$ and $q$ must verify $p-q/\sqrt{p} = \Omega(\sqrt{\log n/n})$ since otherwise at least one node will have more edges towards the other community (see [2, 30]). The fitted curve has a better asymptotic because of the variance of our experiments. Second, the two experimental curves of Louvain and our slight modification essentially coincide: making the assumption that Louvain uses swap is therefore a fair assumption to make as it simplifies the proof greatly.

## 6 Broader Impact

We give the first theoretical explanation of Louvain's success. We show that Louvain not only recovers the hidden partition in the stochastic block model successfully, but also does so in linear time and so for a large range of parameters. Interestingly, if Louvain is properly seeded it can recover the parameters nearly up to the information theoretic threshold.

As explained in the introduction, the goal of this paper is to cast a new light on the success of a popular heuristic for clustering, namely LOUVAIN. With more than 10 000 citations, LOUVAIN is the method of choice for graph clustering. Thus, explaining its power and limitation is of primary importance for a large variety of research areas (see for instance Hoffman et al. [22], analyzing the Bible with LOUVAIN, or Wu et al. [33] for drug repositioning). Our work shows that for graphs exhibiting a clear but noisy clustering structure, then Louvain quickly converges to a global optimum (w.r.t. the modularity objective). Therefore, when the clusters maximizing modularity align

with the ground-truth clusters, Louvain is indeed a powerful clustering algorithms with a reliable performance.

Finally, our work also improves the theoretical analysis and provides tools for a wide-range of other algorithms including Kernighan-Lin, Majorty and other combinatorial algorithms that rely on moving nodes to communities to which they have the most number of edges. Concretely, we show that the probability for a node to have more edges towards its own community is $1/2 + \Omega(\min(\Delta(p - q)/\sqrt{np}, 1))$ in the SBM$(2n, p, q)$, where $\Delta$ is the imbalance. Note that this bound is asymptotically tight. In addition, we also develop strong combinatorial methods that despite dependent variables $(read - 2)$ allow us to analyze a vast amount of cuts. These insights are important for many combinatorial algorithms.

## Acknowledgments and Disclosure of Funding

Ce projet a bénéficié d'une aide de l'État gérée par l'Agence Nationale de la Recherche au titre du Programme Appel à projets générique JCJC 2018 portant la référence suivante : ANR-18-CE40-0004-01.

The work of Vincent Cohen-Addad has been partially done while at CNRS.

This work was supported in part by NSF Award Numbers CF-1461559, CCF-0939370, and CCF-1810758 when Frederik Mallmann-Trenn was affiliated with MIT.

## Footnotes

[1] A balanced cut is a set of edges whose removal splits the graph into two components with equal number of vertices.

[2]Clearly, there are dependencies between the nodes due to the shared edges and one can't apply a standard Chernoff bound to derive the probability of $\mathcal{P}$ to be satisfied. However, we argue that the dependencies are weak, allowing us to use Theorem A.1 to obtain concentration bounds on the probability of the the above mentioned property $\mathcal{P}$ being true. The obtained probability of $\mathcal{P}$ not holding is exponentially small in $\Delta(p-q)$.

[3]In principle, we could avoid introducing the random variables $Z_j$ and only work with $Y_j$, but to avoid some dependency issues, we chose to introduce the fresh variables $Z_j$s.

[4]For the analysis of MAJORITY we obtain the better bound $(p - q)/\sqrt{p} = \Omega(1/n^{1/4})$ since we can simply use $L^* = 1$.

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
