[Supplementary Material]

# On the Power of Louvain for Graph Clustering
# Supplementary Material

## A   The Stochastic Block Model and Definitions

In the following, we use $X \sim \mathcal{P}$ to denote that the random variable $X$ follows the law $\mathcal{P}$. For any integer $n$ and $1 \geq p, q \geq 0, p > q$, we write $\mathrm{SBM}(n, p, q)$ to denote the Stochastic Block Model on $2n$ vertices where the vertices belonging to equisized clusters $V_1, V_2$ unknown to the algorithm.

A graph $G = (V_1 \cup V_2, E)$ is generated from $\mathrm{SBM}(n, p, q)$ by drawing each edge $(u, v) \in E$ independently from the following distribution

$$\mathbb{P}\left[\,(u, v) \in E\,\right] = \begin{cases} p & \text{if } u, v \in V_i, u \neq v \\ q & \text{if } u \in V_i \text{ and } v \in V_j, j \neq i \\ 0 & \text{if } u = v \end{cases}.$$

See Figure 1a for an illustration of such a graph generated by the Stochastic Block Model.

Interestingly, Newman [31] has shown that the modularity objective is the maximum likelihood of a variant of the SBM on two communities, with prescribed degree distribution.

However, it leaves the natural question open of whether the Louvain heuristic on the SBM with two communities indeed converges to this solution (the hidden partition). Proving the convergence shows that Louvain's local decisions can indeed be powerful enough to reach the global optimum.

### A.1   Further Notation

Let $G$ be a graph generated from $\mathrm{SBM}(n, p, q)$. Let $V$ be the vertices of $G$, and $(V_1, V_2)$ be the ground truth partition of $V$. For any $V_i$, we refer vertices of $V_i$ as vertices or elements of *community* $i$. For any vertex $u$ of community $i$, we let $\mathrm{COMMUNITY}(u) = i$.

For any $i \in \{1, 2\}$, for any partition of the vertices of $G$ into 2 parts $(P_1, P_2)$, we refer to the part that contains the larger number of vertices of community $i$ as the *home* of $i$ and we refer to this part as $\mathrm{HOME}(i)$. Namely, $\mathrm{HOME}\,1 = \mathrm{argmax}_{U \in \{P_1, P_2\}} |U \cap V_1|$ and $\mathrm{HOME}\,2 = \mathrm{argmax}_{U \in \{P_1, P_2\}} |U \cap V_2|$.

We will now make use of the following definition, which plays a central role on the analysis of the algorithms. We say that the partition of the vertices of $G$ into 2 parts has *imbalance* $\Delta$ if for each community $i$, $\mathrm{HOME}(i)$ is such that $|V_i \cap \mathrm{HOME}(i)| \geq n/2 + \Delta$ for $j \neq i$ Given a partition, we use $P(u)$ to denote the part of the partition containing vertex $u$.

For any vertex $v$, we let $N(v)$ denote the set of neighbors of $v$. Consider any partition $(P_1, P_2)$ such that $P_i$ is the $\mathrm{HOME}$ for community $i$. We call the *swap value* of a vertex $v \in P_i$ the quantity

$$Q_{v,j} = |N(u) \cap P_j| - |N(u) \cap P_i| + \frac{\deg(u)}{2m}(\deg(P_i) - \deg(P_j))$$

We call in the following the *main part* of the swap value to be $|N(u) \cap P_j| - |N(u) \cap P_i|$, and the *Louvain part* $\mathrm{L}(u) = \frac{\deg(u)}{2m}(\deg(P_{old} \setminus \{u\}) - \deg(P_{new}))$.

We also define $\mathcal{E}_{\deg}$ to be the event that for $G \sim \mathrm{SBM}(n, p, q)$ the degree of each vertex is in the interval $\left[\frac{1}{2}n(p + q), \frac{3}{2}n(p + q)\right]$.

From Chernoff bounds, this event happens with high probability, and we condition on its success from now on. We will also use throughout the paper the following theorem, that gives a Chernoff-like bound for slightly correlated variables:

**Theorem A.1** ([18]).  *We say a family $Y_1, ..., Y_r$ of indicator variables is* read-$k$ *if there exists a sequence $X_1, ..., X_m$ of independent variables and a sequence $S_1, ..., S_r$ of subsets of $\{1, ..., m\}$ such that*

- $Y_i$ *is a function of* $\{X_j, j \in S_i\}$

- *no element of* $\{1, ..., m\}$ *appears in more than* $k$ *of the* $S_i$*'s*

*For any sequence of read-$k$ variables we have that* $\mathbb{P}\left[Y_1 + ... + Y_r > \mathbb{E}\left[Y_1 + ... + Y_r\right] + \varepsilon r\right]$ *and* $\mathbb{P}\left[Y_1 + ... + Y_r < \mathbb{E}\left[Y_1 + ... + Y_r\right] - \varepsilon r\right]$ *are bounded by* $\exp(-2\varepsilon^2 r/k)$.

# B Proofs of Section 3

*Proof of Lemma 3.2.* We aim at bounding the sum of swap values of vertices in $S_0$. We start by looking at the main term of the swap values, namely $|N(u) \cap V \setminus S| - |N(u) \cap S|$ for a vertex of $S$. We define two types of edges for the edges attached to the vertices of $S_0$. The *type-1* edges are the edges with one extremity in $S_0$ and the other in $V \setminus S$. The *type-2* edges are the edges with one extremity in $S_0$ and the other in $S \setminus S_0$. The *type-3* edges are the edges with both extremities in $S_0$. Thus, for a given vertex $u \in S_0$, the sum of majority swap values is given by the number of type-1 edges $e_1(u)$ adjacent to $u$ minus the number of type-2 edges $e_2(u)$ adjacent to $u$, minus the number of type-3 edges $e_3(u)$ adjacent to $u$. Moreover, for $i \in \{1, 2, 3\}$, let $e_i = \sum_{u \in S_0} e_i(u)$ and let $\Sigma^M_{S_0}$ denote the sum of main term of swap values of the vertices in $S_0$, namely $\Sigma^M_{S_0} = e_1 - e_2 - 2e_3$. Hence,

$$\mathbb{E}[e_1] = |S_0|\left(\left(\frac{n}{2} - \Delta\right)p + \left(\frac{n}{2} + \Delta\right)q\right);$$

and

$$\mathbb{E}[e_2] = |S_0|\left(\left(\frac{n}{2} + \Delta - |S_0|\right)p + \left(\frac{n}{2} - \Delta\right)q\right)$$
$$= |S_0|\left(\left(\frac{n}{2} - \Delta\right)\left(\frac{2p}{3} + q\right) + 2\Delta p\right),$$

since $|S_0| = \frac{1}{3}\left(\frac{n}{2} - \Delta\right)$, and

$$\mathbb{E}[e_3] = \frac{|S_0|(|S_0| - 1)p}{2}.$$

Therefore,

$$\mathbb{E}\left[\Sigma^M_{S_0}\right] = -2|S_0|(\Delta(p - q) + p).$$

We now show that $e_1, e_2, e_3$ are concentrated by applying a standard multiplicative Chernoff bound. We first show that type-3 edges are concentrated.

We show that the $2e_3 > |S_0|^2 p - |S_0|\Delta(p - q)/3$ with probability at least $1 - \exp(-5(n/2 - \Delta)\log n)$. First observe that if $|S_0|^2 < |S_0|\Delta(p - q)/3$ the statement holds trivially. Thus, assume that $|S_0| > \Delta(p - q)/3$. Then, let

$$\delta = \frac{\Delta(p - q)}{3|S_0|p}.$$

We have that

$$\delta|S_0|^2 p = \delta\mathbb{E}[2e_3] = |S_0|\Delta(p - q)/3.$$

Hence, we need to bound $\mathbb{P}[2e_3 > (1 + \delta)\mathbb{E}[2e_3]]$. Assume first that $\Delta(p - q) < 3|S_0|p$ and so standard a multiplicative Chernoff bound gives

$$\mathbb{P}[2e_3 < (1 - \delta)\mathbb{E}[2e_3]] \leq \exp(-\frac{1}{2}\delta^2\mathbb{E}[e_3])$$
$$\leq \exp(-\frac{\Delta^2(p - q)^2}{72p})$$
$$\leq \exp(-5(n/2 - \Delta)\log n),$$

where we have used

$$\frac{p - q}{\sqrt{p}} \geq 100\frac{\sqrt{\log n}}{\sqrt{\Delta}}\max\left(1, \frac{\sqrt{(n/2 - \Delta)}}{\sqrt{\Delta}}\right).$$

Similarly, if $\Delta(p-q) \geq 3|S_0|p$, then

$$\mathbb{P}\left[2e_3 < (1-\delta)\mathbb{E}\left[2e_3\right]\right] \leq \exp(-\frac{1}{2}\delta\mathbb{E}\left[e_3\right])$$

$$\leq \exp(-\frac{\Delta(p-q)|S_0|}{24})$$

$$\leq \exp\left(-\frac{\Delta(p-q)(n/2-\Delta)}{72}\right).$$

Notice that the assumption $\frac{p-q}{\sqrt{p}} \geq 100\sqrt{\log n/\Delta}$ implies $p - q \geq 100\sqrt{p\log n/\Delta} \geq 100\sqrt{\log n/\Delta}\frac{p-q}{\sqrt{p}}$ (using $p \geq p - q$), and so $p - q \geq 10000\log n/\Delta$. Pluging this into the right-hand-side of the inequality yields $\mathbb{P}\left[2e_3 < (1-\delta)\mathbb{E}\left[2e_3\right]\right] \leq \exp(-10000\log n|S_0|)$.

We now turn to show concentration bounds on $e_1$ and $e_2$. Let

$$\mu = \mathbb{E}\left[e_3\right] = |S_0|\left((\frac{n}{2}-\Delta)(\frac{2p}{3}+q) + 2\Delta p\right)$$

and $\delta = \frac{\Delta(p-q)|S_0|}{3\mu}$. We now apply a multiplicative Chernoff bound and obtain, when $\delta < 1$,

$$\mathbb{P}\left[e_2 < (1-\delta)\mathbb{E}\left[e_2\right]\right]$$

$$\leq \exp(-\frac{1}{2}\delta^2\mathbb{E}\left[e_2\right])$$

$$\leq \exp(-\frac{1}{72}\frac{\Delta^2(p-q)^2|S_0|^2}{\mu})$$

$$\leq \exp(-\frac{1}{72}\frac{\Delta^2(p-q)^2|S_0|}{((\frac{n}{2}-\Delta)(\frac{2p}{3}+q)+2\Delta p)})$$

$$\leq \exp(-\frac{1}{216}\frac{\Delta^2(p-q)^2(n/2-\Delta)}{((\frac{n}{2}-\Delta)(\frac{2p}{3}+q)+2\Delta p)})$$

$$\leq \exp(-\frac{1}{432}\frac{\Delta^2(p-q)^2(n/2-\Delta)}{2p\max((\frac{n}{2}-\Delta),\Delta)}).$$

$$\leq \exp(-\frac{1}{432}\frac{\Delta^2(p-q)^2(n/2-\Delta)}{2p\max((\frac{n}{2}-\Delta),\Delta)})$$

$$\leq \exp(-5(n/2-\Delta)\log n).$$

where the last line is obtained by using

$$\frac{p-q}{\sqrt{p}} \geq 100\frac{\sqrt{\log n}}{\sqrt{\Delta}}\max\left(1, \frac{\sqrt{(n/2-\Delta)}}{\sqrt{\Delta}}\right).$$

The concentration bound for $e_1$ is identical.

To conclude, observe that if $e_2 > \mathbb{E}\left[e_2\right] - |S_0|\Delta(p-q)/3$ $e_3 > \mathbb{E}\left[e_3\right] - |S_0|\Delta(p-q)/3$ and $e_1 < \mathbb{E}\left[e_1\right] + |S_0|\Delta(p-q)/3$, then we have that

$$\Sigma_{S_0}^M < -|S_0|(\Delta(p-q)). \tag{3}$$

By the above inequalities and a standard application of union bound, this happens with probability at least $1 - 3\exp(-5(n/2-\Delta)\log n)$.

We can now turn to bounding the sum of the Louvain swap values for vertices in $S_0$, namely

$$\Sigma_{S_0}^L = (2m)^{-1}\sum_{v\in S_0}\deg(v)(\sum_{u\in S}\deg(u) - \sum_{u\in V\setminus S}\deg(u)).$$

Using $\mathcal{E}_{\deg}$, it holds that $m^{-1} \leq (1/2n^2(p+q))^{-1}$ and that $\sum_{v\in S_0}\deg(v) \leq 3|S_0|/2n(p+q)$, hence $(2m)^{-1}\sum_{v\in S_0}\deg(v) \leq 3|S_0|/(2n)$. It remains to bound $D = (\sum_{u\in S}\deg(u) - $

$\sum_{u \in V \setminus S} \deg(u))$. We write $\deg(S) = \sum_{u \in S} \deg(u)$, and bound the deviation of $\deg(S)$ from its expectation as for $\Sigma_{S_0}^M$, by defining two types of edges: let $e_1'$ be the number of edges with two extremities in $S$, $e_2'$ the number of ones with one extremity in $S$ and the other in $V \setminus S$. It holds that $\deg(S) = 2e_1' + e_2'$.

We start by analyzing $e_1'$. $\mathbb{E}\,[\,e_1'\,] = ((n/2 - \Delta)^2 + (n/2 + \Delta)^2)p + (n/2 - \Delta)(n/2 + \Delta)q = n^2/2(p + q/2) + \Delta^2(2p - q)$. Note that $\mathbb{E}\,[\,e_1'\,] \in [n^2p/2, n^2p]$. Let $\delta = \frac{\Delta(p-q)}{10np}$. In the case where $\delta \leq 1$, the multiplicative Chernoff bound yields:

$$
\begin{aligned}
\mathbb{P}\,[\,|\mathbb{E}\,[\,e_1'\,] - e_1'| \leq \delta n^2 p\,] &\geq \mathbb{P}\,[\,|\mathbb{E}\,[\,e_1'\,] - e_1'| \leq \delta \mathbb{E}\,[\,e_1'\,]\,] \\
&\geq 1 - \exp(-\delta^2 \mathbb{E}\,[\,e_1'\,]/3) \\
&\geq 1 - \exp(-\delta^2 n^2 p/6) \\
&\geq 1 - \exp\left(-\frac{\Delta^2(p-q)^2}{600p}\right) \\
&\geq 1 - \exp(-5(n/2 - \Delta)\log n)
\end{aligned}
$$

where the last inequality uses the assumption on $\frac{p-q}{\sqrt{p}}$.

In the case where $\delta > 1$, we have $\Delta(p - q) > np$ and so

$$
\begin{aligned}
\mathbb{P}\,[\,|\mathbb{E}\,[\,e_1'\,] - e_1'| \leq \delta n^2 p\,] &\geq 1 - \exp(-\delta n^2 p/6) \\
&\geq 1 - \exp(-\frac{\Delta(p-q)n}{60p}) \\
&\geq 1 - \exp(-5n \log n)
\end{aligned}
$$

using as before $p - q \geq 10000 \log n / \Delta$.

Similarly, $\mathbb{E}\,[\,e_2'\,] = ((n/2 - \Delta)^2 + (n/2 + \Delta)^2)q + 2(n/2 - \Delta)(n/2 + \Delta)p = n^2/2(p + q) - 2\Delta^2(p - q)$. With the same argument,

$$
\mathbb{P}\,[\,|\mathbb{E}\,[\,e_2'\,] - e_2'| \leq \delta n^2 p\,] \geq 1 - \exp(-5(n/2 - \Delta)\log n).
$$

We can now bound $\deg(S) - \deg(V \setminus S)$. Note that $\mathbb{E}\,[\,|N(V \setminus S)|\,] = \mathbb{E}\,[\,S\,]$, since $|S| = |V \setminus S|$. Hence, combining the two previous bounds with an union bound,

$$
\begin{aligned}
\mathbb{P}\,[\,|\deg(S) - \deg(V \setminus S)| \leq 2\delta n^2 p\,] &\geq \\
&\geq 1 - 2\exp(-5(n/2 - \Delta)\log n).
\end{aligned}
$$

Therefore, the whole sum of Louvain swap value is with probability $1 - 2\exp(-10(n/2 - \Delta)\log n)$ bounded by

$$
\left|\Sigma_{S_0}^L\right| \leq \frac{3|S_0|}{2n} \cdot 2\delta n^2 p = \frac{3|S_0|\Delta(p - q)}{10}. \tag{4}
$$

Adding the sum of majority swap values (Equation 3) with the sum of Louvain swap values (Equation 4), we get that the sum of swap values of vertices in $S_0$ is $\Sigma_{S_0}^M + \Sigma_{S_0}^L \geq -\frac{|S_0|\Delta(p-q)}{2}$. This concludes the first statement of the lemma.

The proof of the second statement is similar to the previous one, and is skipped. $\qquad \square$

*Proof of Lemma 3.1.* Consider a cut $S, V \setminus S$ with imbalance $\Delta$ together with a set $S_0$ of size $(n/2 - \Delta)/3$ of bad vertices where each such vertex has positive swap value. Lemma 3.2 implies that the probability for $S_0$ to have total swap value greater than $0 > -((n/2 - \Delta)/3)\Delta(p - q)$ is at most $3\exp(-5(n/2 - \Delta)\log n)$.

We now bound the probability that there exists at least one such set. Taking a union bound over all subsets of size $(n/2 - \Delta)/3$, we have that the probability that such a set $S_0$ exists is at most

$$
\begin{aligned}
3\exp(-5(n/2 - \Delta)\log n) \cdot \exp(\frac{(n/2 - \Delta)}{3}\log n) \\
\leq 3\exp(-4(n/2 - \Delta)\log n).
\end{aligned}
$$

Hence, with probability at least $1 - 3\exp(-4(n/2 - \Delta)\log n)$, the number of bad vertices with positive swap value is at most $(n/2 - \Delta)/3$. By applying the second statement of Lemma 3.2, we get that there cannot be a set of of size $(n/2 - \Delta)/3$ containign good vertices with sum of swap values negative, with the exact same probability bound. Since there are at least $n/2 - \Delta$ good vertices, the lemma follows. $\qquad\square$

*Proof of Theorem 1.1.* Lemma 3.1 implies that for a random cut with imbalance $\Delta$, with probability at least $1 - \exp(-4(n/2 - \Delta)\log n)$ the number of bad vertices with positive swap values is at most $(n/2 - \Delta)/3$ and the number of good vertices with positive swap values is at least $2(n/2 - \Delta)/3$. Define a cut satisfying such a property to be an *improving cut*. Thus, for any improving cut the algorithm swaps a good vertex with probability at least 2/3.

By union bound over all cuts with imbalance $\Delta$ we can bound the probability that there exists a non-improving cut. Indeed, the number of cuts with imbalance $\Delta$ is

$$\binom{n}{n/2 + \Delta} \cdot \binom{n}{n/2 - \Delta} = \binom{n}{n/2 + \Delta}^2$$
$$\leq (n^{n/2 - \Delta})^2 = \exp(2(n/2 - \Delta)\log n)$$

By Union bound, the probability that there exists one non-improving cut with imbalance $\Delta$ is therefore at most $\exp(2(n/2 - \Delta)\log n) \cdot \exp(-4(n/2 - \Delta)\log n) = \exp(-2(n/2 - \Delta)\log n)$. Now, making another Union bound for all $\Delta$ such that $\frac{p-q}{\sqrt{p}} \geq 100\frac{\sqrt{\log n}}{\sqrt{\Delta}} \max\left(1, \frac{\sqrt{(n/2 - \Delta)}}{\sqrt{\Delta}}\right)$ holds, the probability that there exist a non improving cut is at most $\sum_{\Delta=1}^{n/2-1} \exp(-2(n/2 - \Delta)\log n) \leq 2/n^2$.

Note that the cut with $\Delta = n/2$ is not improving, but has no positive swap and is therefore a local optimum.

Let $\zeta$ be the smallest value for $\Delta$ such that $\frac{p-q}{\sqrt{p}} \geq 100\frac{\sqrt{\log n}}{\sqrt{\Delta}}\max\left(1, \frac{\sqrt{(n/2-\Delta)}}{\sqrt{\Delta}}\right)$ holds. Recall that, by assumption we start with $\Delta \geq 2\zeta$. Putting everything together, we have that for every cut with $\Delta \geq \zeta$ the following holds. The probability of a swap to be good is $p' \geq 2/3$ meaning that the size of the imbalance increases by 1. We assume that with the remaining probability the swap is bad meaning it decreases the imbalance.

By Proposition E.2, we start with $\Delta \geq \zeta + s$. Therefore, we can model this as a biased random walk on the integers starting at $z$ and absorbing states $0$ and $n$. The probability to increase for any state $\notin \{0, b\}$ is given by $p'$ and the probability to decrease is at most $1 - p'$. Thus, by Proposition E.2 parameters $p = p'$, $s = \zeta$, $b = n$ we have that

$$\mathbb{P}\left[Z_T = 0\right] = \frac{\left(\frac{1-p}{p}\right)^b - \left(\frac{1-p}{p}\right)^s}{\left(\frac{1-p}{p}\right)^b - 1} \leq \left(\frac{1-p}{p}\right)^s,$$

Where we used that $\frac{x-y}{x-1} \leq y$ for all $0 < x < 1$, $y \leq 1$. Thus the probability of success, that is reaching imbalance of $n$ is at least $1 - \left(\frac{1-p}{p}\right)^s \geq 1 - 1/2^\zeta \geq 1 - 1/n^2$. By Proposition E.2, the expected convergence time is linear in $n$.

$\qquad\square$

*Proof of Theorem 1.2.* Initializing the algorithm with a random cut and repeating $\log n$ times to boost the probability, the conditions of Lemma 4.7 are met with probability $1 - 1/n$. Hence, with Union bound, the conclusions of Lemma 4.7 hold with probability $1 - O(1/n)$. Therefore, we have both $\frac{p-q}{\sqrt{p}} \geq 200n^{-1/6+\varepsilon}$ (by assumption) and $\Delta \geq n/\log^2 n$, which implies $\frac{p-q}{\sqrt{p}} \geq 200\frac{\sqrt{\log n}}{\sqrt{\Delta}}\max\left(1, \frac{\sqrt{(n/2-\Delta)}}{\sqrt{\Delta}}\right)$. Theorem 1.1 concludes therefore the proof of convergence.

For the convergence time, note that SuppMat D describes how the algorithm can be implemented using $O(\deg(u))$ time to update node $u$ for all nodes $u \in V$. Note that for our regime of $p - q$, all nodes have degrees that are w.h.p. concentrated around $n(p + q)$. Hence the update time per node is $O(n(p + q))$. Furthermore, as argued before, the imbalance performs a biased random and after $O(n)$ swaps of nodes the imbalance will have reached $n$, meaning that we recovered the hidden partition. Thus the total convergence time is $O(n^2(p + q)) = O(m)$. $\qquad\square$

## C  Proofs of Section 4

*Proof of Lemma 4.1.* By Lemma 4.2, Lemma 4.3 and triangle inequality we have

$$\left| \mathbb{P}\left[ X_i(u) \geq \max_{j \neq i} X_j(u) + L_i \right] - \mathbb{P}\left[ Z_i(u) > Z_j(u) \right] \right| \leq$$

$$\leq \frac{\mathrm{L}^*}{2\sqrt{\mathrm{Var}\left[ Y_2 \right]}} + 4\sqrt{\frac{2}{np(1-p)}}.$$

From this inequality and Lemma 4.4, the claim follows using that $\mathrm{Var}\left[ Y_2 \right] \geq (n/2 - \Delta)p(1 - p)$. $\qquad\square$

*Proof of Lemma 4.2.* Fix an arbitrary $X_i$. We will apply Essen's inequality (Theorem E.1). Recall that is the sum two binomials allowing us to write $X_i = \sum_{j=1}^{n/2+\Delta} B_j + \sum_{j=1}^{n/2-\Delta} B_j'$ where $B_j \sim \mathrm{B}(p)$ and $B_j' \sim \mathrm{B}(q)$. Note that $\mu_3(B_1)/\mu_2(B_1) = \frac{p(1-p)(1-2p+2p^2)}{p(1-p)} \leq 1$. Similarly, $\mu_3(B_1')/\mu_2(B_1') \leq 1$. Note that $\sigma^2 \geq \sum_i \mathrm{Var}\left[ B_i \right] \geq \frac{n}{2}p(1 - p)$. Thus, by Theorem E.1, for all $x$ we have

$$\left| \mathbb{P}\left[ X_i \leq x + \mathrm{L}^* \right] - \mathbb{P}\left[ Y_i \leq x + \mathrm{L}^* \right] \right| \leq 2\sqrt{\frac{2}{np(1-p)}}. \qquad (5)$$

We can now prove the lemma. Let $\delta = 2\sqrt{\frac{2}{np(1-p)}}$. We have

$$\mathbb{P}\left[ X_1 \geq X_2 + \mathrm{L}^* \right] =$$
$$= \int_x \mathbb{P}\left[ X_1 = x + \mathrm{L}^* \right] \mathbb{P}\left[ X_2 \leq x \right] dx$$
$$\geq \int_x \mathbb{P}\left[ X_1 = x + \mathrm{L}^* \right] (\mathbb{P}\left[ Y_2 \leq x \right] - \delta) dx$$
$$\geq -\delta + \int_x \mathbb{P}\left[ X_1 = x + \mathrm{L}^* \right] \mathbb{P}\left[ Y_2 \leq x \right]$$
$$= -\delta + \mathbb{P}\left[ X_1 \geq Y_2 + \mathrm{L}^* \right],$$

Similarly,

$$\mathbb{P}\left[ X_1 \geq Y_2 + \mathrm{L}^* \right] =$$
$$= \int_x \mathbb{P}\left[ Y_2 = x \right] \mathbb{P}\left[ X_1 \geq x + \mathrm{L}^* \right] dx$$
$$\geq \int_x \mathbb{P}\left[ Y_2 = x \right] (\mathbb{P}\left[ Y_1 \geq x + \mathrm{L}^* \right] - \delta) dx$$
$$\geq \mathbb{P}\left[ Y_1 \geq Y_2 + \mathrm{L}^* \right] - \delta.$$

Putting everything together yields the first part of the claim

$$\mathbb{P}\left[X_1 \geq X_2 + \mathrm{L}^*\right] - \mathbb{P}\left[Y_1 \geq Y_2 + \mathrm{L}^*\right] \leq 2\sqrt{\frac{2}{np(1-p)}}.$$

By using (5) in the other direction, we can also show that

$$\mathbb{P}\left[Y_1 \geq Y_2 + \mathrm{L}^*\right] - \mathbb{P}\left[X_1 \geq X_2 + \mathrm{L}^*\right] \leq 2\sqrt{\frac{2}{np(1-p)}}$$

which concludes the proof.

$\square$

*Proof of Lemma 4.3.* We show that there exists a coupling such that

$$\forall j, L \leq \mathrm{L}^* : \ \mathbb{P}\left[Y_j + L = Z_j\right] \geq 1 - \frac{L}{2\sigma_2}.$$

Note that $Y_i$ follows the same distribution as $Z_i$. We will rewrite $Y_i + L$ as a variable $Y' \sim \mathcal{N}(\mathbb{E}\left[Z_i\right] + L, \mathrm{Var}\left[Z_i\right])$.

We can bound the total variation distance as follows. For any $L \leq \mathrm{L}^*$

$$\mathrm{TV}(Y_j + L, Z_j) =$$
$$= 1/2\sqrt{\frac{\mathrm{Var}\left[Z_i\right]}{\mathrm{Var}\left[Z_i\right]} - 1 + \frac{(\mathbb{E}\left[Z_i\right] + L - \mathbb{E}\left[Z_i\right])^2}{\mathrm{Var}\left[Z_i\right]} + 0}$$
$$= \frac{1}{2}\sqrt{L^2/\sigma_2^2} \leq \frac{\mathrm{L}^*}{2\sigma_2}.$$

We can couple $Y_j + L$ with $Z_j$ w.p. at least $1 - \mathrm{TV}(Y_j + L, Z_j) \geq 1 - \frac{\mathrm{L}^*}{2\sigma_2}$. $\square$

*Proof of Lemma 4.4.* Let $Z' \sim Z_2$ and let $D$ be given by $D \sim \mathcal{N}(\mu_D, \sigma_D^2)$, with $\mu_D = 2\Delta(p-q)$. and $\sigma_D^2 = 2\Delta(p(1-p) - q(1-q))$. We can rewrite $Z_1$ as $Z_1 = Z' + D$.

Let $f(x) = \frac{1}{\sqrt{2\pi}\sqrt{\mathrm{Var}[Z']}}\exp\left(-\frac{1}{2}\left(\frac{\mathbb{E}[Z']-x}{\sqrt{\mathrm{Var}[Z']}}\right)^2\right)$ be the the PDF of $Z'$ (and $Z_2$ therefore). Let $R = \max\{Z', Z_2\}$. Note that, due to symmetry, $\mathbb{P}\left[Z' = R\right] = 1/2$. Hence,

$$\mathbb{P}\left[Z_1 > Z_2 \wedge Z' = R\right] = \mathbb{P}\left[Z' = R \wedge D \geq 0\right]$$
$$\geq 1/2 - \mathbb{P}\left[D < 0\right].$$

Note that, using $\sigma_D^2 \leq \mu_D$ by Chebychev's inequality with $k = \frac{\mu_D}{\sigma_D}$, $\mathbb{P}\left[D < 0\right] \leq \mathbb{P}\left[D < \mu_D - \sigma_D k\right] \leq \frac{1}{k^2} \leq \frac{\sigma_D^2}{\mu_D^2} \leq \frac{1}{\mu_D}$. Let $y = \mu_D$ and $\sigma = \sqrt{\mathrm{Var}\left[Z'\right]} = \sqrt{\mathrm{Var}\left[Z_2\right]}$.

$$\mathbb{P}[Z_1 > Z_2 \wedge Z' \neq R] = \mathbb{P}[Z_1 > Z_2 \wedge Z_2 = R]$$
$$= \int \mathbb{P}\left[Z_1 > Z_2 \wedge Z_2 = R \mid Z_2 = r\right] f(r)\,dr$$
$$> \int_{\mu}^{\mu+\sigma} \mathbb{P}\left[Z' + D > r\right] \cdot \mathbb{P}\left[Z' \in (r-y,r)\right] f(r)\,dr$$
$$> \int_{\mu}^{\mu+\sigma} \mathbb{P}\left[D > y\right] \cdot \mathbb{P}\left[Z' \in (r-y,r)\right] f(r)\,dr$$
$$= \mathbb{P}\left[D > y\right] \cdot \int_{\mu}^{\mu+\sigma} \mathbb{P}\left[Z' \in (r-y,r)\right] f(r)\,dr.$$

We now bound these terms separately. By symmetry of the normal distribution, $\mathbb{P}\left[\, D > y\,\right] \geq \frac{1}{2}$. By using the symmetry of normal distributions and the 68-95-99.7 rule, we get $\int_{\mu}^{\mu+\sigma} f(r)\, dr \geq (1 - 0.69)/2 > 0.15$. Thus, together with monotonicity of the normal distribution for $x > \mu$, we get for $y \leq \sigma$,

$$\int_{\mu}^{\mu+\sigma} \mathbb{P}\left[\, Z' \in (r - y, r)\,\right] f(r)dr \geq$$

$$\geq \int_{\mu}^{\mu+\sigma} y \min_{r' \in [r-y,r]} f(r')f(r)dr$$

$$\geq y \min_{r' \in [\mu-y,\mu+\sigma-y]} f(r') \int_{\mu}^{\mu+\sigma} f(r)dr$$

$$> 0.15yf(\mu + \sigma) \geq 0.15\frac{y}{\sqrt{2\pi}\sigma}e^{-1/2}.$$

For $y \geq \sigma$ we get

$$\int_{\mu}^{\mu+\sigma} \mathbb{P}\left[\, Z' \in (r - y, r)\,\right] f(r)\, dr \geq$$

$$\geq \sigma\mathbb{P}\left[\, Z' \in (\mu - y, \mu)\,\right] f(\mu + \sigma)$$

$$\geq \sigma\mathbb{P}\left[\, Z' \in (\mu - \sigma, \mu)\,\right] \frac{1}{\sqrt{2\pi}\sigma}e^{-1/2}$$

$$\geq \frac{0.15}{\sqrt{2\pi}}e^{-1/2}.$$

Putting everything together and using that $\sigma = \sqrt{\mathrm{Var}\left[\, Z'\,\right]} \leq \sqrt{np(1 - p)}$ yields

$$\mathbb{P}\left[\, Z_1 > Z_2 \wedge Z' \neq R\,\right] >$$

$$> \mathbb{P}\left[\, D > y\,\right] \cdot \int_{\mu}^{\mu+\sigma} \mathbb{P}\left[\, Z' \in (r - y, r)\,\right] f(r)\, dr$$

$$> \frac{1}{2} \cdot \frac{0.15}{\sqrt{2\pi}}e^{-1/2} \cdot \min\left\{\frac{y}{\sigma}, 1\right\}$$

$$\geq 0.018 \min\left\{\frac{\Delta(p - q)}{\sqrt{np(1 - p)}}, 1\right\}.$$

The claim follows by using the law of total probabilities:

$$\mathbb{P}\left[\, Z_1 > Z_2\,\right] =$$

$$= \mathbb{P}[Z_1 > Z_2 \wedge Z' = R] + \mathbb{P}\left[\, Z_1 > Z_2 \wedge Z' \neq R\,\right]$$

$$\geq 1/2 + 0.018 \cdot \min\left\{\frac{\Delta(p - q)}{\sqrt{np(1 - p)}}, 1\right\} - \frac{1}{2\Delta(p - q)}.$$

$\square$

*Proof of Lemma 4.5.* Recall that $\mathrm{L}(u) = \frac{\deg(u)}{2m}(\deg(P_{old}) - \deg(P_{new}))$.

By $\mathcal{E}_{\deg}$, $\deg(u) \in \left[\frac{1}{2}(n(p + q)), \frac{3}{2}(n(p + q))\right]$. The number of edges $m$ verifies therefore $m \in [n \cdot (n(p + q)), 3n \cdot (n(p + q))]$.

We now bound the degree $\deg(P)$ of a community $P$ of size $n$ with imbalance $\Delta$. The expected value of $\deg(P)$ is $n^2(p + q)$, and this variable is the sum of $2n^2$ indicator variables. However, they are not independent: the variables corresponding to edges between vertices of $P$ appear twice in the sum. We define *type-1* edges as edges with two extremities in $P$, and *type-2* the ones with only one extremity in $P$. We write $\deg(P) = 2E_1 + E_2$, where $E_i$ is the set of type-$i$ edges.

We use the Chernoff inequality to show the concentration of each of these variables. $\mathbb{E}\left[E_1\right] = (n/2 - \Delta)(n/2 + \Delta)q + (n/2 - \Delta)^2 p + (n/2 + \Delta)^2 p = (1 + o(1))n^2(p/2 + q/4)$ Let $\delta = \frac{\Delta(p-q)}{100n(p/2+q/4)}$, so that $\delta\mathbb{E}\left[E_1\right] = (1 + o(1))n\Delta(p - q)/100$. By the multiplicative Chernoff bound,

$$\mathbb{P}\left[|E_1 - \mathbb{E}\left[E_1\right]| \leq n\Delta(p - q)/100\right] \geq$$
$$\geq 1 - \exp(-\delta^2 \mathbb{E}\left[E_i^d\right]/3)$$
$$= 1 - \exp(-(1 + o(1))\delta n\Delta(p - q)/300)$$
$$= 1 - \exp(-(1 + o(1))\frac{\Delta^2(p - q)^2}{30000(p + q)})$$

Similarly,

$$\mathbb{P}\left[|E_2 - \mathbb{E}\left[E_2\right]| \leq n\Delta(p - q)/100\right] \geq$$
$$\geq 1 - \exp(-(1 + o(1))\frac{\Delta^2(p - q)^2}{30000(p + q)}),$$

Combining those equations with triangular inequality, we get that with probability $1 - 3\exp(-(1 + o(1))\frac{\Delta^2(p-q)^2}{30000p})$ it holds that

$$|\deg(P) - n^2(p + q)| \leq 4n\Delta(p - q)/100.$$

Using the equations we have on every term of $L(u)$, we get that with probability $1 - 2\exp(-(1 + o(1))\frac{\Delta^2(p-q)^2}{30000p})$

$$L(u) = \frac{\deg(u)}{2m}(\deg(P_{old}) - \deg(P_{new}))$$
$$\leq \frac{3/2n(p + q)}{1/2n^2(p + q)}4n\Delta(p - q)/10 \leq 24\Delta(p - q)/100.$$

Taking a Union bound over the $2n$ vertices concludes the lemma.

Using Chernoff bound with $\delta' = 1/\sqrt{n(p/2 + q/4)}$, we have (using $\delta'\mathbb{E}\left[E_1\right] \leq n\sqrt{np}$):

$$\mathbb{P}\left[|E_1 - \mathbb{E}\left[E_1\right]| \leq n\sqrt{np}\right] \geq$$
$$\geq 1 - \exp(-\delta'^2 \mathbb{E}\left[E_i^d\right]/3)$$
$$= 1 - \exp(-\frac{(1 + o(1))n^2(p/2 + q/4)}{3n(p/2 + q/4)})$$
$$= 1 - \exp(-(1 + o(1))n/3)$$

Similarly,

$$\mathbb{P}\left[|E_2 - \mathbb{E}\left[E_2\right]| \leq n\sqrt{np}\right] \geq 1 - \exp(-(1 + o(1))n/3),$$

Hence, with probability at least $1 - 2\exp(-(1 + o(1))n/3)$,

$$|\deg(P) - n^2(p + q)| \leq 2n\sqrt{np}.$$

And so

$$L(u) = \frac{\deg(u)}{2m}(\deg(P_{old}) - \deg(P_{new}))$$
$$\leq \frac{3/2n(p + q)}{1/2n^2(p + q)}2n\sqrt{np} \leq 6\sqrt{np}$$

$\square$

*Proof of Lemma 4.6.* Using Lemma 4.1 and Theorem A.1, one can bound the number of good and bad vertices with positive swap value. This gives directly the claimed bounds.

Assume first that $\Delta(p-q)/\sqrt{n} \leq 1$, and consider first the vertices that are not in their home. From Lemma 4.1, the probability that a such a vertex has positive swap value is

$$1/2 + 0.018 \cdot \min\left\{\frac{\Delta(p-q)}{\sqrt{np(1-p)}}, 1\right\} - \frac{1}{2\Delta(p-q)} -$$

$$- \frac{L^*}{2\sqrt{(n/2-\Delta)p(1-p)}} - 4\sqrt{\frac{2}{np(1-p)}}.$$

By assumption on $\Delta, p, q$, it holds that $\frac{\Delta(p-q)}{\sqrt{np(1-p)}} \geq 100n^{-1/6+\varepsilon}$. Moreover, $\frac{1}{2\Delta(p-q)} = \frac{\sqrt{p}}{2\sqrt{np}(p-q)} \leq \frac{n^{-1/3-\varepsilon}}{100\sqrt{p}} \leq n^{-1/6}/100$ (using $\sqrt{p} \geq (p-q)/\sqrt{p}$).

From Lemma 4.5, we get that $L^* \leq \Delta(p-q)/100$, hence $\frac{L^*}{2\sqrt{(n/2-\Delta)p(1-p)}} \leq \frac{\Delta(p-q)}{100\sqrt{(n)p(1-p)}}$ (using $\Delta(p-q) \leq \sqrt{np}$, which implies $n/2 - \Delta \geq n/4$).

Finally, $4\sqrt{\frac{2}{np(1-p)}} \leq \frac{\Delta(p-q)}{100\sqrt{np(1-p)}}$.

Combining all these equations simplifies the probability that a given vertex not in its home is positive to $1/2 + 0.009\frac{\Delta(p-q)}{\sqrt{np(1-p)}}$.

Hence, the expectation of the number of good positive vertices is $\mu \geq n/2 + 0.009\sqrt{n}\Delta(p-q)$. Since these variables are read-2, Theorem A.1 gives a concentration inequality: the expected number good vertices with positive swap value is $\mu \pm \varepsilon n$ with probability $1 - \exp(-\varepsilon^2 n)$ taking $\varepsilon = \frac{\Delta(p-q)}{10\sqrt{(n)}}$ gives that with probability $1 - \exp\left(-\frac{\Delta^2(p-q)^2}{100}\right)$ the number good vertices with positive swap value is $\mu \pm \frac{\sqrt{n}\Delta(p-q)}{10} \geq n/2 + \Theta\left(\sqrt{n}\Delta(p-q)\right)$.

By the same argument, with probability $1 - \exp\left(-\frac{\Delta^2(p-q)^2}{100}\right)$ the number of bad vertices with positive swap value is less than $n/2 - \Theta\left(\sqrt{n}\Delta(p-q)\right)$.

Therefore, with probability $1 - \exp\left(-\frac{\Delta^2(p-q)^2}{100}\right)$ we have

$$\mathbb{P}\left[\,good \mid positive\,\right] = 1/2 + c_1\frac{\Delta(p-q)}{\sqrt{n}},$$

for some constant $c_1$.

In the case where $\Delta(p-q)/\sqrt{n} \geq 1$, Lemma 4.1 gives that the expected number of good positive swaps is $n/2 + nc_2$, for some constant $c_2$. We can therefore take $\varepsilon = c_2/2$ in Theorem A.1 to have the claimed bound. $\qquad\square$

*Proof of Lemma 4.7.* At the beginning, $\Delta_0 = \sqrt{n}$. Let $\Delta_k = n^{1/2+\varepsilon k}$. We show that going from $\Delta_k$ to $\Delta_{k+1}$ happens with probability $1 - O(1/n)$ within $\frac{\sqrt{n}}{p-q}\log n$ steps. We consider first the case where $\Delta_k(p-q)/\sqrt{n} \leq 1$. We base our reasoning on a result proved in [15] (Chapter XIV.2, XIV.3), that gives probability for a biaised random on $\mathbb{N}$ walk to double before being divided by 2.

1. Assume first that Lemma 4.6 holds for enough steps, meaning that we have $\mathbb{P}\left[\,good \mid positive\,\right] = 1/2 + c_1\frac{\Delta(p-q)}{\sqrt{np}}$. With Corollary E.3, this ensures that within $O(\frac{\sqrt{np}}{p-q}\log n)$ steps $\Delta$ is doubled with probability $1 - O(1/n^2)$, and that the imbalance never drops below $\Delta_k/2$.

2. Starting from $\Delta_k$, repeating (1) $r = \log n^\varepsilon$ times ensures to double the imbalance $\log n^\varepsilon$ times, and hence to reach an imbalance $\Delta_k \cdot 2^{\log n^\varepsilon} = \Delta_{k+1}$ and the number $t$ of steps is,

with a Union bound, $\sum_{i=0}^{\log n^\varepsilon} \frac{\sqrt{np}\log^2 n}{p-q}$ with probability $1 - \varepsilon \log n/n^2$. This term sums up to $t = \frac{\sqrt{np}\log^3 n}{p-q} \le 2n^{2/3+\varepsilon}\log^3 n$.

3. We prove now that we can indeed assume that Lemma 4.6 holds long enough. The number of cuts that the algorithm can reach after $t$ steps is $2^{t\log n}$. Over all these steps, we know that the imbalance is bigger than $\Delta_k/2$, and therefore at every fixed step $\mathbb{P}\left[\,good \mid positive\,\right] = 1/2 + c_1 \frac{\Delta_k(p-q)}{2\sqrt{np}}$ with probability $1 - \exp\left(-\frac{\Delta_k^2(p-q)^2}{100p}\right) = 1 - \exp\left(-\frac{n^{2/3+2(k+1)\varepsilon}}{100p}\right)$. Since $2^{t\log n} = 2^{n^{2/3}\log n}$ with probability $1 - O(1/n^2)$, we can simply take an union bound over all these $t$ steps, and therefore, with probability at least $1 - 1/n$ the new imbalance is $\Delta_{k+1}$ (the bottleneck in the probabilities being the time to double $\Delta$).

4. To reach a $k$ such that either $\Delta_k = O(n)$ or $\Delta_k(p-q)/\sqrt{n} \ge 1$, repeating $O(1/\varepsilon)$ times these three steps is enough. This is constant, and therefore it is possible to take another Union bound on the probability of (3) to ensure that all the repetitions run through with probability $1 - O(1/n)$.

Hence, with high probability, it holds that after $O(1/\varepsilon \cdot \log n^\varepsilon \cdot \frac{\sqrt{np}}{p-q}) = O(\frac{\sqrt{np}\log n}{p-q})$ steps, the imbalance $\Delta$ verifies $\frac{\Delta(p-q)}{\sqrt{np}} = 1$. Hence, $\Delta = \Omega(\frac{\sqrt{np}}{p-q}) = \Omega(n^{2/3-\varepsilon})$.

We now turn to the case where $\Delta_k(p-q)/\sqrt{n} \ge 1$. In that scenario, Lemma 4.6 gives that, with probability $1 - \exp(-c_2^2 n/4)$, $\mathbb{P}\left[\,good \mid positive\,\right] = 1/2 + c_2$. Using Corollary E.4, the expected waiting time to double the imbalance is therefore $O(\Delta)$, and so the time to reach an imbalance $\frac{n}{100c_2^2\log^2 n}$ is with high probability $\frac{n}{100c_2^2\log n}$. Since the probability $\mathbb{P}\left[\,good \mid positive\,\right]$ of Lemma 4.6 holds with probability $1 - \exp\left(-\frac{c_2^2 n}{4}\right)$ it is possible to make a union bound on the $\frac{n}{100c_2^2\log n}$ steps of the algorithm. $\qquad\square$

# D   An Efficient Implementation

In this section we show how Louvain can be implemented so that the cost for swapping node $u$ is simply $O(\deg(u))$ for a const number of communities. In particular our results on the SBM imply that the total convergence time is $O(m)$, which is asymptotically optimal. The core idea is to maintain for every node $u$ and every part $P_j$ the modularity (Equation 1) if we moved $u$ to $P_j$. Our implementation relies on keeping the following data for community $i$:

1. $D_i$ : The sum of degrees of nodes in $P_i$, i.e., $D_i = \sum_{u\in P_i}\deg(u)$.

2. $D_i^*$ : The sum of degree pairs of nodes in $P_i$ scaled by $1/(2m)$, i.e., $D_i^* = \sum_{u,v\in P_i}\deg(u)\cdot\deg(v)/(2m)$.

3. For all $x \in \{-n, n\}$: $L_{i,j}[x]$: A list of nodes in $P_i$ with swap value of $x$ w.r.t. $P_j$.

When moving a node $u$ to $P_i$, it's easy to see that updating $D_i$ can be done in constant time. Furthermore, using $D_i$ one can easily update $D_i^*$ in constant time. Finally, updating $L_{i,j}[x]$ is possible in time $O(\deg(u))$, by considering each neighbor $v \in P_i$. Let $x_v$ be the swap value of $v$ w.r.t. to the $P_i$ and $P_j$. If $(u,v) \notin G$, then $x_v$ does not change and hence $v$ remains in $L_{i,j}[x_v]$. Otherwise $x_v$ decreases by either one or two depending on whether $u \in P_j$ or some other $P_\ell$. Therefore it suffices to move $v$ to either $L_{i,j}[x_v - 1]$ or $L_{i,j}[x_v - 2]$. This move can be implemented using linked lists so that it takes $O(1)$ operations.

Using the above implementation is it easy to sample a node u.a.r. from the nodes with positive swap value between $P_i$ and $P_j$: Determine the smallest $x_{min}$ such that $x_{min} \ge D_i^*$. Then sample $u$ from all $\cup_{x\ge x_{min}} L_{i,j}[x]$ uniformly at random. The sampling step only takes $O(1)$ time.

# E   Auxiliary Claims

In this section, we show that the assumption of making *swaps*, i.e.: enforcing that the algorithm keeps a balanced partition, instead of allowing to move a vertex from one side to the other if it improves the modularity, is *without loss of generality*. Namely, we show that the bounds claimed in Theorems 1.1, 1.2 also apply to the latter variant of the algorithm. We show that the size imbalance between the two parts times $p$ remains small compared to $\Delta(p-q)$ throughout the execution of the algorithm, and so that this imbalance has a negligible role in the execution of the algorithm.

The following is a slightly weaker version of Theorem 1 in [6].

**Theorem E.1** (Esseen inequality [6]). *Let $\mu_k(X)$ denote the kth absolute central moment $\mu_k = \int |x - \mathbb{E}[X]|^k \mathbb{P}[X = x]\, dx$. Let $X_1, \ldots, X_n$ be a collection of $n$ random variables, with $\mu_2(X_i) > 0$ for all $i$. Let $\mu = \mathbb{E}[X]$ and $\sigma^2 = \sum_i \mathrm{Var}[X_i]$. Let $F(\cdot)$ be the commutative density function (cdf) of $X = \sum_i X_i$ and let $G(\cdot)$ be the cdf of $\mathcal{N}(\mu, \sigma^2)$. Then,*

$$\sup_{-\infty < x < \infty} |F(x) - G(x)| \leq \frac{1.88}{\sigma} \max_i \frac{\mu_3(X_i)}{\mu_2(X_i)}.$$

**Proposition E.2** (Chapter XIV.2, XIV.3 in [15]). *Let $p \in (0,1) \setminus \{1/2\}$ and $b, s \in \mathbb{N}$. Consider a discrete time Markov chain $(Z_t)_{t \geq 0}$ with state space $\Omega = [0, b]$ where*

- $Z_0 = s \in [0, b]$

- $\mathbb{P}[Z_t = i \mid Z_{t-1} = i - 1] = p$ *for* $i \in [1, b-1], t \geq 1$

- $\mathbb{P}[Z_t = i \mid Z_{t-1} = i + 1] = 1 - p$ *for* $i \in [1, b-1], t \geq 1$

- $\mathbb{P}[Z_t = i \mid Z_{t-1} = i] = 1$ *for* $i \in \{0, b\}, t \geq 1$

*Let $T = \min\{t \geq 0 \mid Z_t \in \{0, b\}\}$. Then,*

$$\mathbb{P}[Z_T = b] = \frac{\left(\frac{1-p}{p}\right)^s - 1}{\left(\frac{1-p}{p}\right)^b - 1} \quad \text{and}$$

$$\mathbb{P}[Z_T = 0] = \frac{\left(\frac{1-p}{p}\right)^b - \left(\frac{1-p}{p}\right)^s}{\left(\frac{1-p}{p}\right)^b - 1}.$$

*Moreover,*

$$\mathbb{E}[T] = \frac{s}{1 - 2p} - \frac{b}{1 - 2p} \cdot \frac{1 - \left(\frac{1-p}{p}\right)^s}{1 - \left(\frac{1-p}{p}\right)^b}.$$

**Corollary E.3.** *Let $X_n$ be a random walk on the integer line, such that $\forall t, \mathbb{P}[X_{t+1} = X_t + 1] = 1/2 + c_1 \frac{\Delta(p-q)}{2\sqrt{np}}$ if $X_t \neq \Delta/2, 2\Delta$ and $X_{t+1} = X_t$ with probability 1 otherwise. Then, if $X_0 = \Delta$, $\Delta \geq \sqrt{n}$ and $\frac{p-q}{\sqrt{p}} \geq n^{-1/6}$, it holds with probability at least $1 - 1/n^2$ that for $T = \frac{\sqrt{np}}{p-q} \log^2 n$, $X_T = 2\Delta$.*

*Proof.* We use Proposition E.2 with $s = \Delta/2, b = 3\Delta/2, p = 1/2 + c_1 \frac{\Delta(p-q)}{2\sqrt{np}}$ and $Z_t = X_t - \Delta/2$.

Remark that $\frac{1-p}{p} = \frac{1/2 - c_1 \frac{\Delta(p-q)}{2\sqrt{np}}}{1/2 + c_1 \frac{\Delta(p-q)}{2\sqrt{np}}} = 1 - x$ with $x = \frac{2c_1 \frac{\Delta(p-q)}{2\sqrt{np}}}{1/2 + c_1 \frac{\Delta(p-q)}{2\sqrt{np}}} \geq \frac{\Delta(p-q)}{\sqrt{np}}$.

We have:

$$\mathbb{P}\left[\,Z_T = 0\,\right] = \frac{\left(\frac{1-p}{p}\right)^b - \left(\frac{1-p}{p}\right)^s}{\left(\frac{1-p}{p}\right)^b - 1}$$

$$\leq \frac{(1-x)^s}{1 - (1-x)^b}$$

$$\leq 2\,(1-x)^s \leq 2\exp(-sx)$$

$$\leq 2\exp(-\frac{\Delta^2(p-q)}{\sqrt{np}})$$

$$\leq 2\exp(-n^{1/3})$$

where the last inequality uses $\Delta \geq \sqrt{n}$ and $\frac{p-q}{\sqrt{p}} \geq n^{-1/6}$. Moreover, the expected waiting time to reach $T$ is

$$\mathbb{E}\left[\,T\,\right] = \frac{s}{1-2p} - \frac{b}{1-2p} \cdot \frac{1 - \left(\frac{1-p}{p}\right)^s}{1 - \left(\frac{1-p}{p}\right)^b}$$

$$\leq 2\frac{b}{2p-1} \leq 2\frac{3\Delta/2}{c_1\frac{\Delta(p-q)}{2\sqrt{np}}}$$

$$\leq 6\frac{\sqrt{np}}{c_1(p-q)}$$

Hence, by Markov inequality, it holds with probability $1 - O(1/n^2)$ that $T \leq \frac{\sqrt{np}}{(p-q)}\log^2 n$. This concludes the lemma. $\qquad\square$

**Corollary E.4.** *Let $X_n$ be a random walk on the integer line, such that $\forall t, \mathbb{P}\left[\,X_{t+1} = X_t + 1\,\right] = 1/2 + c_2$ if $X_t \neq \Delta/2, 2\Delta$ and $X_{t+1} = X_t$ with probability 1 otherwise. Then, if $X_0 = \Delta$ and $\Delta \geq \sqrt{n}$, it holds with probability at least $1 - 1/n^2$ that for $T = \Delta\log^2 n$ $X_T = 2\Delta$.*

*Proof.* We use Proposition E.2 with $s = \Delta/2$, $b = 3\Delta/2$, $p = 1/2 + c_2$ and $Z_t = X_t - \Delta/2$. We have $\frac{1-p}{p} = 1 - x$ with $x = \frac{2c_2}{1/2+c_2} \geq 2c_2$. As in Corollary E.3, $\mathbb{P}\left[\,Z_T = 0\,\right] \leq 2\exp(-sx) \leq 2\exp(-\Delta c_2) \leq 2\exp(-c_2\sqrt{n})$. Moreover, $\mathbb{E}\left[\,T\,\right] \leq \frac{2b}{2p-1} = \frac{3\Delta}{2c_2}$. Hence, by Markov's inequality, $\mathbb{E}\left[\,T\,\right] \leq \Delta\log^2 n$ with probability $1 - 1/n^2$. $\qquad\square$

# F    Allowing Size Imbalances

In this section we explain how to deal with different cluster sizes. Let $V$ be the ground truth and let $P = (P_1, P_2)$ the current partition. Let $\mu = \frac{|P_2|-|P_1|}{2}$ be a parameter that controls the difference in cluster size, and $\Delta$ such that $|P_2 \cap V_2| = n - \mu/2 + \Delta$ (and therefore $|P_1 \cap V_2| = n + \mu/2 - \Delta$, $|P_2 \cap V_1| = n - \mu/2 - \Delta$ and $|P_1 \cap V_2| = n + \mu/2 + \Delta$).

The expected value of a good swap is therefore $2\Delta(p-q) - \mu(p+q)$. We argue that $\mu(p+q)$ is always negligible compared to $\Delta(p-q)$, which allows to adapt all the technical lemmas.

In particular, for the cold start and Lemma 4.6, the definition of variables $Y_1$ and $Y_2$ changes: $Y_1(u) \sim \mathcal{N}((n/2 + \Delta - \mu/2)p + (n/2 - \Delta - \mu/2)q, \sigma_1)$ and $Y_2(u) \sim \mathcal{N}((n/2 - \Delta + \mu/2)p + (n/2 + \Delta + \mu/2)q, \sigma_2)$. The proof of Lemma 4.2 goes through as before, and the one of Lemma 4.3 is modified to take into account the difference between $Y$ and $Z$. However, their expectations differs by $\mu(p+q)$, hence the lemma statements remains unchanged assuming $\mu(p+q)$ is negligible compared to $\Delta(p-q)$. Lemma 4.6 is unchanged as well, under the same assumption.

The proof of Lemma 4.7 can then be adapted as follows, to take this size imbalance into account.

- The first and second bullet remain valid. We add to them the following observation: Assuming Lemma 4.6 holds for enough steps, $\mu$ follows a random walk biased towards 0: we argue that $\mathbb{P}\left[\mu_{k+1} = \mu_k + 1\right] \leq 1/2 - \mu_k/n$ if $\mu_k \neq 0$. The probability that we pick a node from the smaller side is $\frac{n-2\mu}{2n} = \frac{1}{2} - \frac{\mu}{n}$. The probability to move the smaller side, condition on picking a node from the larger side is at most $1$. Thus, the probability of increasing $\mu_k$ by one is: $\mathbb{P}\left[\mu_{k+1} = \mu_k + 1\right] \leq \left(\frac{1}{2} - \frac{\mu}{n}\right)$. Hence, using a simple random argument, $\mu_k$'s maximum during the first $i$ steps is w.h.p. at most $6\sqrt{i}\log n$.

- The third bullet is rewritten as follows. We prove now that we can indeed assume that Lemma 4.6 holds long enough. The number of cuts that the algorithm can reach after $t$ steps is $2^{t\log n}$. Over all these steps, we know that the imbalance is bigger than $\Delta_k/2$, and moreover that $\mu \leq 6\sqrt{t}\log n \leq 6n^{1/3+\varepsilon/2}\log^{5/2} n$. Since $\Delta_k \geq n^{1/2}$ and $\frac{p-q}{p} \geq n^{-1/6+\varepsilon}$, this implies that $\mu p \leq \frac{\Delta(p-q)}{2000}$ and that the condition of Lemma 4.6 are met. Therefore at every fixed step $\mathbb{P}\left[\,good \mid positive\,\right] = 1/2 + c_1\frac{\Delta_k(p-q)}{2\sqrt{np}}$ with probability $1 - \exp\left(-\frac{\Delta_k^2(p-q)^2}{100p}\right) = 1 - \exp\left(-\frac{n^{2/3+2(k+1)\varepsilon}}{100p}\right)$. Since $2^{t\log n} = 2^{n^{2/3}\log^3 n}$ with probability $1 - O(1/n^2)$, we can simply take an union bound over all these $t$ steps, and therefore, with probability at least $1 - 1/n$ the new imbalance is $\Delta_{k+1}$ (the bottleneck in the probabilities being the time to double $\Delta$).

- The fourth bullet stays alike, as the conclusion of the proof.

The other lemmas for the regime of large $\Delta$ can be adapted using the fact that, since the number of steps is linear, $\mu p = O(\sqrt{n}\log n \cdot p)$ is negligible compared to the swap value $\Delta(p - q) = \Omega(\frac{n^{5/4+\varepsilon}}{\log n} \cdot p)$. The proof of Lemma 3.2 is straightforwardly adapted to that regime.