[Reviews · NeurIPS 2020]

Review 1

Summary and Contributions: After reading the authors' rebuttal, my evaluation remains the same. The submission studies the behaviour of Louvain in the Stochastic Block Model. For the case of two clusters, the submission gives the theoretical analysis of the Louvian algorithm when the underlying graph is generated according to the Stochastic Block Model.

Strengths: The submission presents the first analysis on the approximation result for the Louvian algorithm when the underlying graph is generated from the SBM.

Weaknesses: - All the analysis only holds when the underlying graph has two clusters, and from their analysis it looks that the presented techniques cannot be easily generated to the case of more clusters. - The presentation of the submission isn't good, see my comments below. In particular, several pages of the submission are devoted to sketch their proof; in order to achieve this, the submission lists many notation and technical lemmas. I feel that it's difficult to follow the technical discussions in Section 3 and 4, and with the current form of the presentation it would be difficult for a reader to catch the key ideas behind those technical conditions and relationships among different parameters. - The Section of Experiments is weak. I understand that the entire focus of the submission is about the Louvian algorithm; however, a reader might wonder why they should choose to use the Louvian algorithm instead of the other ones, e.g. spectral clustering, for which the theoretical guarantee is known. Hence, some experimental result comparing your proposed algorithm with other clustering algorithms would provide information on the merit of the proposed algorithm.

Correctness: As stated in the section above, I couldn't check the claims in detail.

Clarity: No, the writeup could be significantly improved.

Relation to Prior Work: Yes, the relation to previous work is clearly discussed.

Reproducibility: Yes

Additional Feedback: - I would suggest you to include the formal definition of the SBM in the paper to make it more self-contained. - Line 83 - 87: what's epsilon here? - Line 170: what is P(u) here? - Line 173: *consists* of successive steps. - Line 178: you mention that the algorithm stops when the partition P is such that P(u)=P_{i*_u}. Do you need to compute i_u* in advance? What's the time complexity to compute these for all vertices? - The paragraph starting with Line 173 discusses the Louvain algorithm, and the L181 states that "the algorithm above, Balanced Louvian,...". So what's Balanced Louvian algorithm? You further discussed three differences from L182, however, it's not clear to me whether Balanced Louvian is a new algorithm proposed in this submission or it's from previous reference. Some clarification is needed. - Line 202: you defined Home1 and Home2 to the parts that contain the highest number of vertices of community 1 and community 2. I couldn't under the definitions here. Some formal definition and further clarification is needed. - Line 290: We will define *two normally* distributed ... - The section of experiments needs to be improved significantly. See my comment on the weakness of the submission.


Review 2

Summary and Contributions: The paper analyzes the balanced Louvain and Majority methods for recovering the bi-partition of graphs generated by the two-block stochastic block model. The authors show that 1) Balanced Louvain can recover the ground-truth bi-partition if O(n) iterations. This result holds in case of warm-starting, meaning that the initial solution has n/2 + Delta overlap with one of the target communities. The authors also assume that an inequality is satisfied that captures the relationship of the model parameter p, q, the imbalance Delta and the number of nodes n in the communities. 2) In case that there is no sufficient overlap in the initial solution (cold-start), then the authors shows that Balanced Louvain will recover the optimal partition in O(n) iterations. What changes compared to the previous result (warm-start) is the assumption on the relation among the model parameter p, q, the imbalance Delta and the number of nodes n in the communities. 3) The authors also improve the analysis for Majority. In particular they show that Majority can recover the ground-truth bi-partition without assuming dense graphs, which was the previous state-of-the-art result. By removing the assumption on the density of the graph the authors show that the running of Majority is sub-quadratic.

Strengths: (+) It's a new analysis using the two-block stochastic block model of a widely used heuristic method for graph partitioning. (+) Improves upon previous state-of-the-art analysis The results are theoretically grounded and novel.

Weaknesses: (-) Only considers the two-block stochastic block model case.

Correctness: Yes

Clarity: No. I think this is the major drawback of this paper. Initially, I was so excited about the paper, but as I was reading I could not get passed it's poor quality. See my comments below on possible ways that this can be improved.

Relation to Prior Work: Yes. The authors discuss extensively the previous state-of-the-art and how they improve upon it. Generally in the paper you mention that local search algorithms are not well analyzed. I think it worths defining what local search methods you have in mind. I know that you mean methods that rely on local operations and Louvain is such a method. However, there is also a large class of methods called "local graph clustering methods" that are rigorously analyzed with wost- and average-case guarantees. These local methods are clearly not heuristics.

Reproducibility: Yes

Additional Feedback: 1) Providing the citation count in the abstract is a bit weird. Saying that Louvain is widely used by practitioners is enough. 2) In the abstract you mention that you utilize the stochastic block model, which is true. However, in the main paper you only use the two-block version of it. This should be clarified in the abstract 3) In the abstract you mention "many other combinatorial algorithms". Please mention the algorithms. 4) Line 19-22. You spend so many lines describing an intermedia results that is used in your theorems and you do not describe the theorems themselves. 5) Line 26: remove "and" from "and Loyd's" 6) Lines 24 - 34. You try to make a connection to gradient descent for neural networks and claim this method is not well analyzed and we do not know its guarantees and running time for converging to local minima. This is not true. The last 5 years there have been numerous papers that do exactly that. I recommend to remove such vague and incorrect statements. 7) Line 63. You mention Louvain's local optima. Please give a definition for these local minima. 8) Line 75. You have not provided the definition of imbalance. 9) Line 85. What is epsilon here? 10) Line 96. You mention running time O(n^2 p), then in line 100 you mention linear running time. Please clarify. 11) Line 121. Replace practice with practical? 12) Line 137. Replace "has" with "that". 13) Line 140. How does your analysis compare with the state-of-the-art analysis on the two-block stochastic block model? 14) Line 170. P(u) is not defined. 15) Lines 169 - 178. This is a very poor description of the algorithm. You need to add some wording/summary. For example, Balanced Louvain swaps two nodes that do not belong in their optimal part of the partition. The algorithms terminates when this is not true etc. 16) Line 181. "This algorithm above", I guess you refer to Balanced Louvain that you described in lines 169 - 178. 17) Where is the caption for Figure 1? 18) Figure 1a is totally redundant. 19) The ticks in the y-axis of Figure 1b are not useful at all to understand the the smallest-value. You can use the actual values for ticks and simply mention that the plot is not log-scale. 20) Line 202. Replace "contains" with "contain" 21) Lines 200 - 202. The description of "Home" is very poor. The one given in the appendix is much more accurate. 22) Lines 203 - 204. I think the word "bad" is confusing. Just by looking the given definition one would think that these nodes are the good nodes. However, I think you call these nodes bad since if one of them has positive swap value, then the algorithm might swap them, and this would be a bad decision. So, either changing the wording or extend the definition so that it makes sense. 23) Line 233. I think you need to assume that S_0 is at least of size (n/2-Delta)/3, and not exactly of that size. 24) Line 237. Similar comment here. 25) Line 244. Maybe you wanted to say "fresh partition" instead of "fresh graph"? 26) Line 347. Replace "observation" with "observations". 27) Line 355. I would not call the the SBM a good representation of real graphs. I would remove the word "practice" from this sentence. Supplementary material 30) Line 478. remove second "we". 31) Line 484. You call this imbalance, but I really think this should be called "sufficient overlap". 32) Lines 480 - 481. The definition of home depends on the ground-truth partition. This sentence needs to be rephrased to reflect that, otherwise it is confusing. ###### ###### ###### ###### ###### ###### ###### I have read the rebuttal, my evaluation remains the same. ###### ###### ###### ###### ###### ###### ######


Review 3

Summary and Contributions: The authors present a simple variant of the Louvain algorithm, proves exact recovery conditions in the two-cluster stochastic block model when the algorithm is carefully or randomly initialized, and provide numerical evidence that the proposed algorithm performs well.

Strengths: The theoretical results of this paper come timely and are valuable to the ML community. They help us better understand local search algorithms for clustering, such as the Louvain algorithm, that perform extremely well in practice but lack theoretical guarantees. Although the algorithm under study is a simplified version of the standard Louvain algorithm, it does help the authors obtain interesting results. The numerical experiments show the similar performance of the simplified version and the standard one, therefore justifying the simplification.

Weaknesses: Thm 1.1 seems loose when the algorithm is randomly initialized. Random equal-sized cluster initialization would result in small \Delta. Yet, a small \Delta (say \Delta=o(n)) would make the bound in Thm 1.1 vacuous. It would be great if the authors could point this out explicitly.

Correctness: The arguments appear correct.

Clarity: Yes.

Relation to Prior Work: The differences from previous work is clearly discussed. I do suggest the authors cite more work on algorithms for SBM such as iterative methods [1-4], SDP [5-9] and spectral algorithms [10, 11], as they have laid the foundation for algorithmic analysis under SBM. [1] Gao, Ma, Zhang, and Zhou. Achieving optimal misclassification proportion in stochastic block model. [2] Lu and Zhou. Statistical and computational guarantees of Lloyd’s algorithm and its variants. [3] Zhang and Zhou. Theoretical and Computational Guarantees of Mean Field Variational Inference for Community Detection. [4] Gao, Ma, Zhang and Zhou. Community Detection in Degree-Corrected Block Models. [5] Bandeira. Random laplacian matrices and convex relaxations. [6] Hajek, Wu and Xu. Achieving exact cluster recovery threshold via semidefinite programming. [7] Guédon and Vershynin. Community detection in sparse networks via Grothendieck’s inequality. [8] Fei and Chen. Exponential error rates of SDP for block models: Beyond Grothendieck’s inequality. [9] Fei and Chen. Achieving the Bayes Error Rate in Synchronization and Block Models by SDP, Robustly. [10] Abbe, Boix, Ralli and Sandon. Graph powering and spectral robustness. [11] Abbe, Fan, Wang and Zhong. Entrywise eigenvector analysis of random matrices with low expected rank.

Reproducibility: Yes

Additional Feedback: Overall, the writing of the paper could be much improved by more proof-reading. Here I provide a non-exhaustive list of suggestions: - Theorem 1.2: \epsilon not defined - L190, bigger strictly -> strictly - L197, we call …: this sentence should appear before L181 when Standard Louvain is introduced for the first time - L256, weak (read-2): what is “read-2”? - L369, theoretical analysis provides -> theoretical analysis that provides - L525 in supp, last line of displayed equation: missed a right parenthesis - L530 in supp, first line of displayed equation: last \leq is redundant --- Post-rebuttal update --- I have read the authors' rebuttal, and it has addressed my concerns.


Review 4

Summary and Contributions: The paper presents a theoretical analysis of the Louvain method for graph clustering. The paper considers a two-community stochastic block model and analyzes the behavior of the Louvain algorithm in this setting. The paper shows, that under certain conditions, the Louvain method recovers the optimal partition and that it converges rapidly. Specifically, the authors show that when initialized with equal-size two-partition and for a certain range of the SBM model parameters, it converges to the optimal partition in O(n) rounds. ----- I thank the authors for their response. I think the paper would greatly improve if the analysis is extended to k > 2.

Strengths: - Louvain is one of the most commonly used techniques for graph clustering, and the paper presents the first theoretical results on the behavior of the Louvain method. - The paper includes some novel results that extends beyond the Louvain method, including a bound on the probability that a node in SBM has more edges toward its own community.

Weaknesses: The main limitation of the work is the relatively narrow setting: the SBM model, the focus on two-community graph clustering, and the use of a modified ('balanced') variant of the Louvain algorithm. This means that this work does not provide theoretical guarantees that are necessarily applicable in real scenarios, and it is unclear if the theoretical tools that are developed are applicable for the study of more general graph clustering settings. In particular, while the use of the balanced Louvain method is theoretically and empirically justified, it is not clear that these results extend beyond the considered scenario (for example, it would be interesting to compare the balanced method and the standard Louvain method, empirically, on real datasets).

Correctness: While I did not verify the proofs in the supplementary material, I did not notice mistakes in the main paper. However, many of the claims in the paper, as well as the empirical results, are limited to the narrow scenario considered in the paper. While this is OK, it needs to be stated clearly (for example, the abstract does not mention the fact that the analysis is limited to two communities).

Clarity: I think the paper is generally well written. See comments below for typos and inconsistencies. In particular, the Louvain term L used in Section 4 is only defined in the supplementary material.

Relation to Prior Work: The paper includes a detailed discussion on previous work and the highlights the contribution of the paper with respect to previous work.

Reproducibility: Yes

Additional Feedback: Minor comments: - The Louvain term L used in Section 4 is only defined in the supplementary material - line 107: redundant period after references - line 124: "missclassifies an tiny" should be "misclassifies a tiny" - line 137: "such has to" - line 155: "some addition notation" - line 218: use of squared brackets in Lemma 3.1, while other Lemmas are using parentheses. - Seems like there is a different notation for set subtraction in line 200 and line 220. - line 320: "We not" -> "we note"

[Author Response · NeurIPS 2020]

We thank the reviewers for the thorough reviews and helpful comments.

- Concerning the generalization to a larger number of communities (**Rev1, 2, 4**):

As common in the majority of research on the SBM, we decided to focus on $k = 2$ as it encapsulates the core-challenges and the main technical contributions. We expect the overall results to generalize to the case of $k > 2$. In the course of the work leading to the current paper, we formally proved the following theorem statement for $k > 2$ for a "parallel" variant of Louvain where all swaps are done "in parallel" (moving at each step every vertex to its best community, disregarding the choices of other vertices): *Let $\varepsilon > 0$, not necessarily constant. Consider a graph $G \sim SBM(k, n, p, q)$ with $k = O(1)$. Then, with probability at least $2/3$, Parallel Louvain recovers the partition $\{V_1, V_2, \ldots, V_k\}$ after $O(\varepsilon^{-1})$ rounds, if $p - q \geq \frac{\log^5 n}{n^{1/4 - \varepsilon}}$.* We expect this statement to extend naturally to standard (sequential) Louvain.

We did not attempt to include such a statement in the submission due to readability concerns (the proofs for $k > 2$ do become longer, but not much more interesting) and due to space limits. Indeed, the purpose of our paper is to prove the first theoretical guarantees on a non-trivial setting for the Louvain algorithm, together with understanding its limitations. We believe that understanding the behavior of Louvain on simple, yet non-trivial, clusterable graphs is a first step towards getting better algorithms. Hence, in our opinion focusing on two communities helps make our argument clearly.

Given the Reviewers' interest, we propose to add further comments on the case of $k > 2$ to the closing remarks of our revised paper.

- Concerning the experimental section (**Rev1**):

We would like to stress that the goal of the paper is to explain the success of the Louvain algorithm – which is widely used, in particular due to its efficiency and easiness to implement – and expose the strength and weaknesses of the modularity objective function. It is well known that semi-definite programming approaches achieve optimal recovery bound in the stochastic block model, and would naturally outperform Louvain in terms of recovery. However, the running time is significantly worse rendering them impractical for even medium-sized graphs. Moreover, spectral approaches are known to be very brittle and often fail in practice when noise is present, and in general we believe Louvain is much more robust. However, before considering such sophisticated clustering scenarios, it is benefitial to understand the standard case (i.e., the stochastic model) and it turns out that this already presents a plethora of challenges that need powerful machinery to be overcome.

Our goal was not to provide an optimal algorithm for the stochastic block model, but to establish the first provable bounds for Louvain, taking one step at a time and laying the groundwork for subsequent extensions.

- Concerning the presentation (**Rev1, 2**):

We acknowledge the typos and presentation comments, and we will address them. Note that $O(n^2 p)$ does indeed signify linear running time (our running time is linear w.r.t. to the input size, which is linear in $n^2 p$).

- Concerning the warm start (**Rev3**):

The point of Thm 1.1 is to show that a 'good' initialization (with high Delta), obtained using e.g. some other algorithm or additional knowledge, allows Louvain to converge to the ground-truth community in a near-optimal regime of $p - q$. We agree that for a random initialization, Thm 1.1 is not useful, in which case we appeal to Thm 1.2 which handles the random initialization scenario but for a stronger assumption on $p - q$. We will make this explicit in the paper.

- Concerning balanceness condition (**Rev4**):

In fact, we actually prove that the balancing step is not necessary. This is done in suppl. mat., Section F: "Allowing Size Imbalances". Note that we did not include this (important) improvement in the main proof for readability (since it seems already fairly sophisticated) and in view of the page limit. In the final version, we'll make sure to emphasize that we do not need the balancing.

[Meta-Review · NeurIPS 2020]

During discussion among reviewers, two concerns are focused and discussed: (1) the proposed formulation is limited to the case of k = 2; (2) experiments are not thorough. The limitation of (1) is a standard assumption for SBM models and it can be acceptable. In addition, the concern (2) is also acceptable as evaluating Louvain itself is not the scope of the paper, and the main contribution of this paper is theoretical analysis of Louvain. Hence I recommend acceptance of the paper. All reviewers agree that presentation can be improved. So I strongly recommend to revise the paper by referring reviews.